# Anticipating the Climate Change Impacts on Madeira's Agriculture: The Characterization and Monitoring of a Vine Agrosystem

**Miguel Â. A. Pinheiro de Carvalho** [1,2,3] **, Carla Ragonezi** [1,2,*]**, Maria Cristina O. Oliveira** [1]**, Fábio Reis** [1]**, Fabrício Lopes Macedo** [1,2]**, José G. R. de Freitas** [1]**, Humberto Nóbrega** [1] **and José Filipe T. Gança** [1]

1    ISOPlexis Centre Sustainable Agriculture and Food Technology, University of Madeira, Campus da Penteada, 9020-105 Funchal, Portugal
2    Centre for the Research and Technology of Agro-Environmental and Biological Sciences (CITAB), University of Trás-os-Montes and Alto Douro, 5000-801 Vila Real, Portugal
3    Faculty of Life Sciences, University of Madeira, Campus da Penteada, 9020-105 Funchal, Portugal
*    Correspondence: carla.ragonezi@staff.uma.pt; Tel.: +351-291-705-002

**Abstract:** Climate—Madeira Strategy (CMS) foresees two models to describe the climate scenarios for the Madeira region in 2050 and 2070. These scenarios anticipate an average temperature rise of 1.4 to 3.7 °C and a decrease in precipitation by 30 to 40%. Consequently, Madeira's agriculture will suffer the impacts of climate change. To understand these impacts, a baseline of major agrosystem components needs to be established, with the ultimate goal to monitor its consequences in its functioning. CASBio project used the 1961–1991 and 2010–2020 meteorological data series to modulate climate conditions and characterize and monitor six agrosystems for 2 years. One of them was a vineyard, *Quinta das Vinhas*, representing a typical agrosystem in the Mediterranean climate. The annual and seasonal variation in climatic parameters, soil conditions, microbiological communities, floristic and insect diversity, and crop production was assessed, using a total of 50 parameters. The results were used to establish a baseline of the agrosystem components and their seasonal and annual variation. The major findings are: (i) winter and summer extreme events show a trend in temperature and precipitation supporting a fast change in climate; (ii) a critical imbalance between nitrogen-fixing and denitrifying bacteria was identified, especially in summer, that could be determined by the rise in temperature and drought; (iii) among floristic diversity, the therophytes and geophytes confirm to be the most suitable indicators for the rise in temperature and reduction in precipitation in the agrosystems; (iv) an imbalance in favor of *C. capitata* plague was observed, associated with the summer rise in temperature and decrease in precipitation; (v) despite an increase in most of the grape varieties production, the Madeiran wine local varieties were shown to be less stable in productivity under observed climate conditions. The agrosystem baseline is a starting point for long-term monitoring and allows for further quantifying the influence of climate change on agrosystem productivity, resilience, and sustainability.

**Keywords:** model; agrosystem indicators; productivity; sustainability; agrodiversity

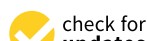



## 1. Introduction

Climate changes are a major driver that will affect all ecological systems and activities in the present and next centuries. These changes are detrimental in different fields, and agriculture and food production are not an exception [1]. The increase in productivity to feed an increasing population, while considering the sustainability of the production systems, is a major challenge. According to Intergovernmental Panel on Climate Change (IPCC) [2], the impacts of these changes could seriously threaten the food production systems security. These losses would be the result of changes in the suitable arable land, the seasons' duration, and the decrease in crop yield potential, imposed by the extreme events

of temperature, drought, water scarcity, diseases, plagues, and loss of crop adaptation to growth conditions. However, climate change impacts are often inferred from models that are general and geographically large scale in scope, while scientists and policymakers are pressured to gather evidence and deal with the consequences locally [3]. Thus, it is important to analyze changes and create small-scale models for local environments, like Madeira islands. Moreover, small islands, due to their geographical isolation, physical characteristics and size, are recognized as being highly vulnerable to these impacts [4].

The IPCC report concludes that temperature has been rising since the middle of the 20th century and is increasing between 1 and 2 °C in the Macaronesia Region [2]. Even these increases in average conditions would result in disproportionate changes in the frequency of extreme or marginal conditions. These long-term processes, operating over tens and hundreds of years, will profoundly affect the soil, vegetation, and communities of animals present and their interconnection into functioning ecosystems [3]. For Madeira, future climate scenarios were developed in the framework of the Climate—Madeira Strategy (CMS), based on IPCC models. They forecast an overall average temperature rise of 1.4 to 3.7 °C and a decrease in precipitation by 30 to 40% by 2070. Specifically, an average rise in temperature from 1.6 to 2.5 °C and a decrease of around 33% in precipitation are expected by 2070, for *Estreito da Calheta*, southwest of Madeira [5], where our case study is located. However, these model-based prognostics, that take other model outputs as inputs, generate a "cascade of uncertainty" [3], even more relevant in small territories. There is a need to monitor, locally and regularly, these ecosystem parameters and extreme events, follow real-world changes, and create or refine local models.

In insular regions, such as the Madeira archipelago, the predicted climate changes could be catastrophic. It can represent a significant loss of agrosystems and territory degradation, with particular focus on soil erosion and loss of its productivity potential. These could create a significant risk for the island's food security, in all its component availability, access, use, and stability [4]. Macedo and Pinheiro de Carvalho [6] simulated climate change for 2070 with data presented by the CMS using the Multicriteria Climate Classification System [7], as well as the Winkler Index (WI). The authors report that, if climate change occurs as predicted for specific areas in Madeira, according to WI, the final quality of the wines will be altered. Therefore, in this work, we propose a strategy to monitor the annual and seasonal variation in climate, soil conditions, microbiological communities, floristic and insect diversity, and crop production. Our goal was to establish the agrosystem baseline, allowing us to follow the predicted climate changes and their impacts in agrosystem functioning. These represent increased concerns about the region's dependency on the external supply chains that will be threatened by future global food crises, as well as the loss of major agricultural export goods from Madeira. This imposes the need to assess the impacts of the changes, dictated by temperature variation, precipitation, and radiation, in regional agriculture, agrosystems, and major crop productions, particularly. The influence of temperature and moisture on the soil parameters, such as carbon content and decomposition, nutrient volatilization [8,9], species loss and turnover [10], crop phenology and yield potential [11], and insect and plague occurrence [8], was already mentioned.

Madeira's agriculture is mainly mountain-type-agriculture developed in a narrow territory, showing mixed Subtropical and Mediterranean climatic elements. Traditional agrosystems (farms), called *fazendas*, mainly have a size below one hectare and are organized in one or more small terraces or field plots (*poios*). They are bounded by volcanic stonewalls, irrigated by water canals (*levadas*), which directly obtain water from upper mountains. The landscapes, agrosystem structure and size determine that agriculture is mainly familiar, based on low input, rotation, intercropping, or pluricultural practices. Crop irrigation is mostly based on traditional rain-fed and flood irrigation systems. Those practices require the availability of water resources that can be reduced in a climate change scenario. The main agricultural productions are fruticulture, including subtropical, e.g., banana, custard apple, and avocado, or temperate ones, e.g., apples, cherries, and grapes vine, and

horticulture, including potatoes, sweet potato, taro, beans, and a wide range of leaf, fruit, and bulb vegetables.

*Calheta* is a Madeiran County with agroecological conditions favoring subtropical or temperate agriculture, characterized by its warm Mediterranean climate, with little predominance of rain and water resources for agricultural purposes, due to weather conditions and landscape orientation.

The *Quinta das Vinhas* (QV), the place chosen to implement the case study, is an agritourism farm that was converted to organic production, starting in 2018, and mainly produces grapes for winemaking. The major regional varieties used for wine production are *Verdelho*, which is predominant, *Malvasia*, and *Sercial*. Exotic varieties, such as *Bastardo* and *Syrah*, are also used for winemaking. Other varieties growing in the farm are *Terrantez*, *Malvasia Fina* (*Boal* or *Bual*), *Malvasia Cândida Roxa*, *Malvasia Cândida*, *Tinta Antiga de Gaula*, and *Tinta Negra Mole*. This agrosystem does not represent a traditional farm system of Madeira, but it is in an area threatened by climate change and holds one of the major agricultural goods produced in the region. The transition for organic framing allows for monitoring and eventually modulates the evolution and impacts of climate scenarios [5] in this production mode. According to early agroclimatic zoning, the expectable change in climatic conditions will increase the arable area suitable for viticulture [12], but crop productivity and quality can suffer from climate extreme events. Thus, a need exists to deeply understand how these changes will influence the varieties' productivity and the quality of the final wine and, further, how the agrosystems' structure and balance between different agrobiodiversity elements will be affected.

Considering the expectable climate changes, and the need to maintain a sustainable food production system, promoted by European Green Deal Pact and Farm to Fork strategy, there is an urgent need for exploratory studies. Despite more than 15 years [2] of climate change assessment and more than 10 years of delineation of CMS [5], which proposed the climate scenario models, no systematic studies to address ongoing changes and their impact on agrosystems have yet been performed. The physical features of the Madeira region and the diversity of agrosystems and crops, do not allow for the easy application of usual computational climate models, or the explanation of ongoing changes and their impacts. The present work is the first systematic study aimed to monitor climate change and its impacts on Madeira's agrosystems. The starting point for our work had to be CMS and its confrontation with observed climate conditions, and the more recent 10-year climate series. We hypothesize that the climate shift is accelerating, and the negative impact on agrosystem functioning and crop development is already significant. Nonetheless, to quantify and understand these impacts, a baseline in the agrosystem conditions needed to be established. To do so, our case study, *Quinta das Vinhas*, was monitored and characterized on various levels, including (i) climatic conditions; (ii) soil physicochemical and biological features, e.g., microbiological indicators; (iii) floristic plant diversity; (iv) insect survey and diversity; (v) crop production, as an attempt to propose a model of ongoing changes in the agrosystem and obtain data that can be used by regional entities in actions to be taken.

This work is part of a wider data collection, generated by the CASBio project—assessment and monitoring of agrobiodiversity and sustainability of agrosystems in the new climate scenarios. This project aims to assess its sustainability in the expected climate scenarios for Madeira Island, and to develop knowledge and technology that can contribute to increase the agriculture resilience and promote the local bioeconomy.

## 2. Materials and Methods

### 2.1. Case Study

The *QV* case study is in *Estreito da Calheta* (Figure 1), with a total and arable land area of 2.37 and 1.99 ha, respectively, at an altitude ranging between 305 and 347 m above sea level. The south-oriented farm has a slightly rugged orography, built-in terraces, and trimmed stone walls, mostly occupied by vineyards. The vines are oriented in espaliers in major parts of the plots, with a small part maintained in branches supported by traditional trellises.

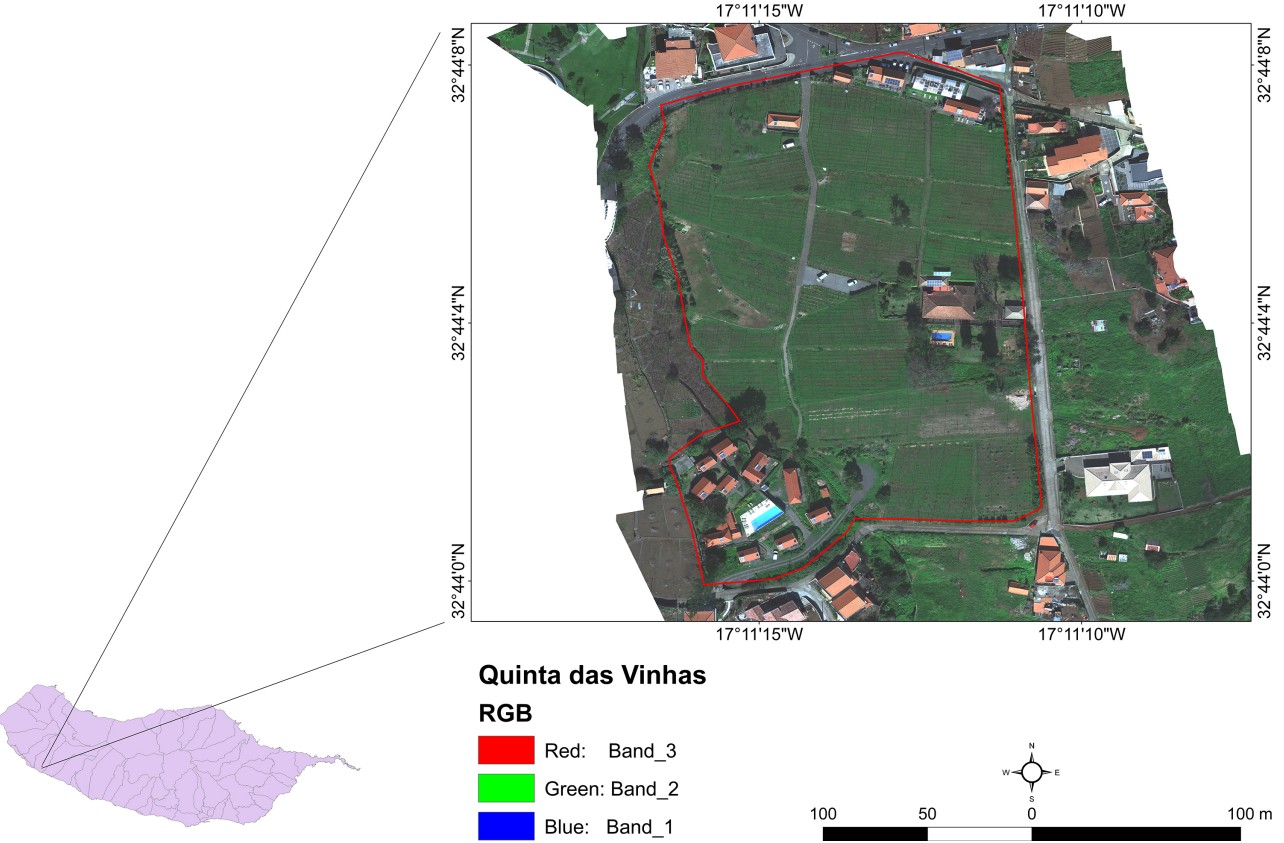

**Figure 1.** Orthophoto map showing the location of the *Quinta das Vinhas* (*QV*) agrosystem case study.

### 2.2. Climate and Climatic Conditions

The climatic conditions for Madeira and QV case study were assessed using data obtained from 11 meteorological stations of the Portuguese Institute for Sea and Atmosphere, I.P. (IPMA, IP) (Table 1), and a meteorological station (Davis Vantage Pro 2 Wireless) installed on site. The periodicity of data collection was monthly. Quarterly data results were obtained from an average of the daily data for the period under study. The climatic variables used were average, maximum, and minimum temperatures (°C), accumulated precipitation (mm), average humidity (%), and global radiation (kJ/m$^2$).

**Table 1.** Meteorological stations were used to assess and model climatic conditions in the Madeira and *Quinta das Vinhas* case study.

| Stations | Owner | Latitude (DD) | Longitude (DD) | Altitude (m) |
|---|---|---|---|---|
| *Funchal/Observatório* | IPMA | 32.65 | −16.89 | 58 |
| *Sanatório do Monte* | IPMA | 32.65 | −16.90 | 380 |
| *Santa Catarina/Airport* | IPMA | 32.69 | −16.77 | 58 |
| *Lugar de Baixo/P. do Sol* | IPMA | 32.68 | −17.09 | 40 |
| *Camacha* | IPMA | 32.66 | −16.83 | 680 |
| *Bom Sucesso* | IPMA | 32.65 | −16.90 | 290 |
| *Chão do Areeiro* | IPMA | 32.72 | −16.92 | 1590 |
| *Ponta Delgada* | IPMA | 32.75 | −16.71 | 133 |
| *Santana* | IPMA | 32.81 | −16.89 | 380 |
| *Bica da Cana* | IPMA | 32.76 | −17.06 | 1560 |
| *Santo da Serra* | IPMA | 32.73 | −16.82 | 660 |
| *Quinta das Vinhas* | ISOPlexis | 32.73 | −17.19 | 337 |

### 2.3. Climate Data Modeling

A set of data from historical series of 30 years (1961–1990) and the most recent 10 years (2010–2020) of monthly average temperature and monthly precipitation for Madeira was used. Data modeling was conducted using ArcGIS software version 10.6.1. [13]. The process methodology and parameters used in the spatialization of temperature were prepared according to Macedo et al. [12], Cavalcanti et al. [14], and Rosa [15]. The interpolation of precipitation data from IPMA weather stations was performed using the Inverse Distance Weighting (IDW) method. Models were created for variation in the average temperature and annual minimum and maximum precipitation observed in the early 30 and last 10 years.

The two future climate scenarios for 2070–2099 (ΔA2, ΔB2) proposed by the Adaptation Strategy for Climate Changes in Madeira Region [5] were used to compare the actual climate in the case study with climate conditions for expected scenarios.

### 2.4. Soil Sampling and Characterization

Soil samples were collected according to the protocol from Paetz et al. 2005 [16], with adaptations. Briefly, soil subsamples were taken at various points in a zigzag along the field terraces, at a depth between 15 and 20 cm. Sub-samples were mixed to create bulk homogeneous samples representative of the local soil. Twelve soil samplings were collected per quarterly field mission, for 2 years (8 missions in total).

In the laboratory, samples were weighed and processed for analysis. Edaphic parameters evaluated were soil temperature and humidity, physicochemical parameters, such as pH, organic matter, macro and micronutrients, and field capacity. Physicochemical parameters were analyzed every six months.

### 2.5. Microbiologic Parameters

The soil's biological features included the analysis of microbiota using classical microbiology and molecular techniques.

The classical microbiology included the account of nitrogen-fixing and denitrifying microorganisms, total bacteria, and total fungi, using the dilution method and inoculation in the appropriate media, mannitol agar (MA), nitrate broth (NB), potato dextrose agar (PDA), and nutrient agar (NA).

The molecular analysis was performed through the total genomic DNA extraction according to Yeates et al. [17]. Microorganism groups (bacteria, fungi, archaea, arbuscular mycorrhizal fungi-AMF) were analyzed by the Terminal Restriction Fragment Length Polymorphism (T-RFLP) method. Polymerase chain reaction (PCR) amplification was performed with the labeled forward primer (5′ 6-FAM). The Bacteria 16S rRNA gene was amplified with the 8F (AGAGTTTGATCCTGGC TCAG)/1489R (TACCTTGTTACGACTTCA) primers. For Fungi, ITS region with ITS1 (TCCGTAGGTGAACCTGCGG)/ITS4 (TCCTCCGCT-TATTGATATGC) primers. Archaea 16S rRNA gene using the 109F (ACKGCTCAGTAA-CACGT)/915R (GTGCTCCCCCGCCAATTCCT) primers. AMF 18S rRNA gene was firstly amplified with NS1 (GTAGTCATATGCTTGTCTC)/NS4 (CTTCCGTCAATTCCTTTAAG) and then with AML1 (ATCAACTTTCGATGGTAGGATAGA)/AML2 (GAACCCAAA-CACTTTGGTTTCC) primers. PCR products were subjected to T-RFLP analysis using 2 restriction enzymes for each group: *Hinf I/Hae III* for bacteria and fungi, *Hae III/Alu I* for Archaea, and *Hinf I/Mbo I* for AMF. Terminal restriction fragments (T-RFs) were determined by Genescan analysis software (Applied Biosystems-Stabvida, Portugal) and analyzed using T-REX software [18] to obtain the matrices. Diversity indices were determined based on T-RFs. Species richness (S), abundance (N), Corrected Evenness (E′), Shannon–Wiener Diversity (H′), and Simpson (D) were calculated using the statistical program Community Ecology Parameter Calculator (ComEcoPaC) version 1.0 [19], according to Nóbrega et al. [20]. The formulas used in the tool are based on Krebs [21], Southwood and Henderson [22], and Tischler [23] and can be checked in *Help* button.

### 2.6. Floristic and Plant Diversity Indicators

The agrosystem floristic assay assessment was carried out using quadrants and transects, accounting for the number of wild plant species specimens [24]. Samplings were quarterly collected for 24 months (8 in total). The following indices, Species richness (S), Shannon–Wiener Diversity (H′), Corrected Evenness (E′), and Hill's Index (N2), were calculated as mentioned in the previous point.

### 2.7. Insect Survey and Diversity Indicators

Insect samplings were quarterly collected for 21 months (7 in total). Two types of traps were used: Fruit fly EOSTRAP® Tephri trap for *Ceratitis capitata* (Wiedemann, 1824), with 3 specific food attractants for females (Econex Trypack Compact) and one insecticide diffuser; apple cider vinegar handmade trap (PET bottle), mainly for insects of the Order Diptera. Eight vinegar traps were placed and remained in the field for 7 days in the first week of each quarter. The two fruit fly traps remained for all 21 months in the field, with the food attractants and the insecticide diffuser being changed every 90 days. The total insect biomass was determined for capture in the vinegar traps. In addition to total biomass, the number of males and females of *C. capitata* was also considered.

### 2.8. Crop and Production Elements

The winegrower from QV provided an inventory of the property's varieties and annual production data per variety. The values obtained are expressed in kg·h$^{-1}$ for the varieties *Bastardo*, *Malvasia*, *Sercial*, *Terrantêz*, *Verdelho*, and *Syrah*.

### 2.9. Data Treatment

All data were statistically analyzed. Data or logarithmic transformed data were tested for normal distribution. When normal distribution was verified, ANOVA followed by the Tukey test were performed to determine the existence of statistically significant differences among the variables. When the data normality was not observed, Kruskal–Wallis and Dunn's tests were applied. Correlation analysis between variables was performed using the Pearson coefficient or when normality was not observed, Spearman's correlation coefficient. Statistical analyses were performed using the statistical package for the social sciences (SPSS) version 23.0 for Windows (IBM Corp, Armonk, NY, USA). Images for microbiology were performed with RStudio Version 1.3.1056, using the function boxplot from Package graphics and theme update from Package ggplot2.

### 3. Results and Discussion

### 3.1. Climate and Climatic Conditions

The Madeira archipelago is in the subtropical region of the Atlantic Ocean and belongs to the Macaronesia biogeographic sub-region, which also includes the archipelagos of the Azores, Selvagens, Canary Islands, and Cape Verde, and a part of the northwest coast of Africa. The archipelago is made up of two main inhabited islands, Madeira (742 km$^2$) and Porto Santo (43 km$^2$), and three non-habited islands (Desertas). According to Cruz et al. [25], the main island has two climate types: Mediterranean and temperate. Further, according to these authors, the annual average temperature is around 19 °C in the coastal zones; however, in the highest points, the temperature can drop around 9 °C. Consequently, Madeira has mainly a temperate Mediterranean climate, with dry conditions, from April to October, particularly on the south coast, where *Quinta das Vinhas* is located.

#### 3.1.1. Evolution of Temperature Variation in Winter and Summer

According to the models accessing future climate changes for 2070–2099, an average rise in the temperature by up to 3.7 °C and a decrease by up to 40% of accumulated precipitation are expected. To better understand ongoing climatic changes, the historical climatic series (1961–1990) used by Gomes et al. [5] was compared to more recent meteorological data (2010–2020) referring to the variation in average temperature and accumulated pre-

cipitation. The spatialization of Madeira Island's temperature was carried out for winter (December to February) and summer (June to August) extreme events (Figure 2A,C). The spatialization comparison of the most recent average temperature variation (Figure 2B,D) with historical ones (Figure 2A,C) shows a shift of 8 °C and more than 6 °C in winter and summer, respectively.

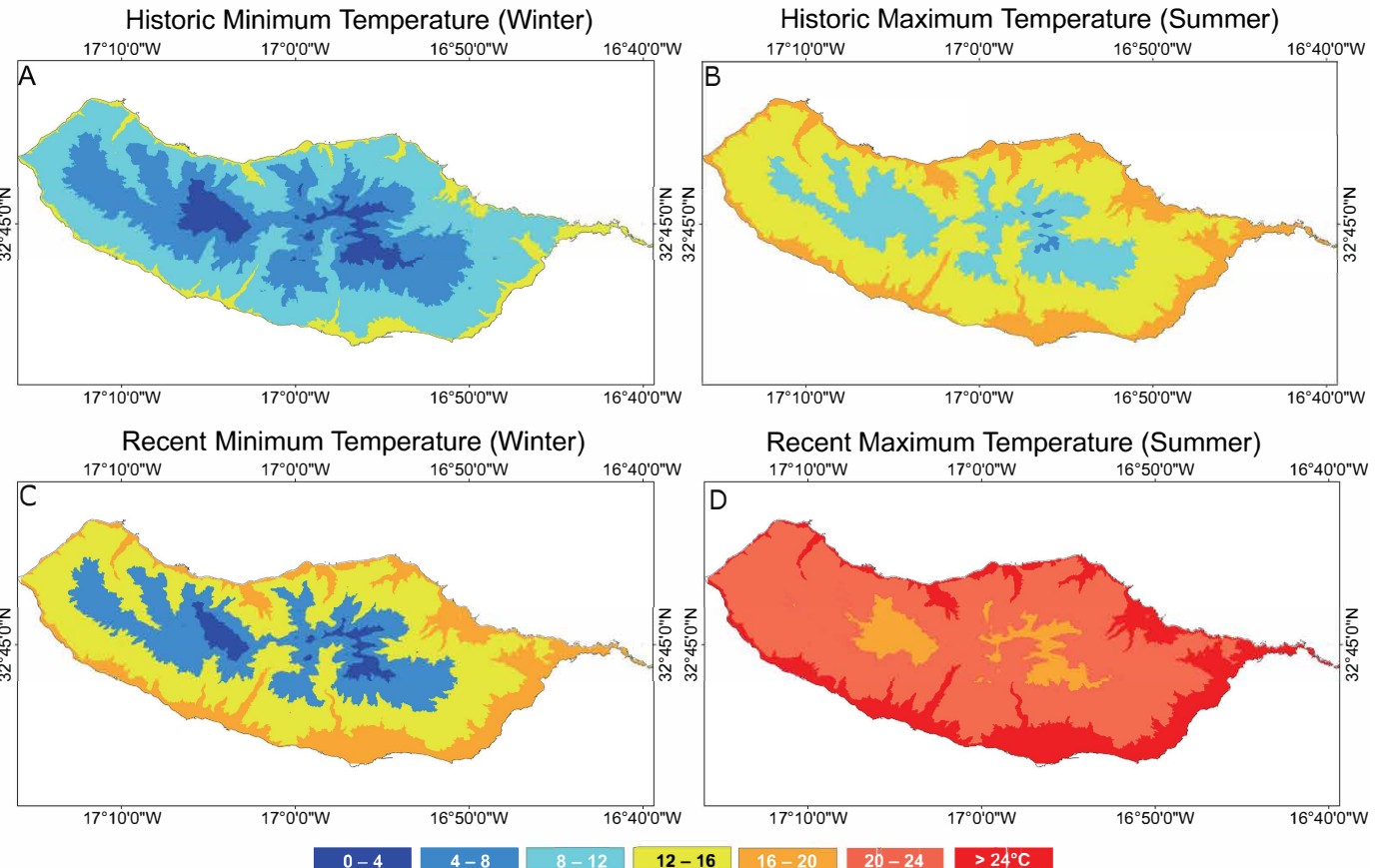

**Figure 2.** Spatialization of the average temperature of Madeira Island in extreme events of summer and winter, using the historical series (1960–1991), maps (**A**,**C**), and recent series (2010–2020) maps (**B**,**D**).

Deep analysis of the results regarding the minimum average temperatures (winter) allowed us to notice a temperature shift, reaching more than 4 °C, with a variation ranging between 12 and 16 or 20 and 24 °C, above or below 300 m on the south and north coast, respectively (Figure 2B). These contrasting changes are more pronounced on the south coast and in the QV location. It is also possible to observe an expansion of the area where higher temperatures occur. Another highlight is the total replacement of the 8 to 12 °C thermic strip observed in Figure 2A (blue light) by two new thermic ranges of 12 to 16 °C and 16 to 20 °C (Figure 2B, yellow and orange colors). This shows that the current winters tend to be warmer, which is correlated with a fast increase in average temperature and temperature amplitudes.

Similar findings were observed when we compare the recent maximum average temperatures in summer with the historical ones (Figure 2C,D). The maximum temperature in summer reached above 24 °C on both coasts, at altitudes below 300 m, on average, more than 4 to 6 °C. These models show that Madeira Island's thermal range in altitude becomes warmer during the summer. These findings are consistent with the increase in the number of tropical days and nights registered in Madeira Island in the last few years.

### 3.1.2. Evolution of Accumulated Precipitations in Winter and Summer

Interpolation of the accumulated precipitation in the winter and summer periods was carried out. The comparison of precipitation results of recent with historical series is given in a similar way to those for the winter and summer average temperatures. The interpolation maps show that the increase in temperature is accompanied by a decrease in accumulated precipitation. In recent years, the winter accumulated precipitation (Figure 3B) shows values ranging between 320 and 640 mm in a major part of the island territory, including on the south coast. Therefore, the accumulated precipitation in the winter period experienced a half reduction in the recent series when compared with the historical ones (Figure 3A). The recent series of summer precipitation range between 0 and 35 mm, on some limited places of the south coast and 160–320 mm range in the northwest and central massif, with precipitation volumes of 80–160 mm in major parts of the island, including in the QV location (Figure 3D). These models show that Madeira Island's climate is becoming dryer in summer, with a drastic half reduction in water resources provided by accumulated precipitation, when compared with the historical series (Figure 3C). These data indicate the emergence of an extremely dry climate, in major areas of the southwest coast of Madeira, with the appearance of severe drought conditions for agriculture.

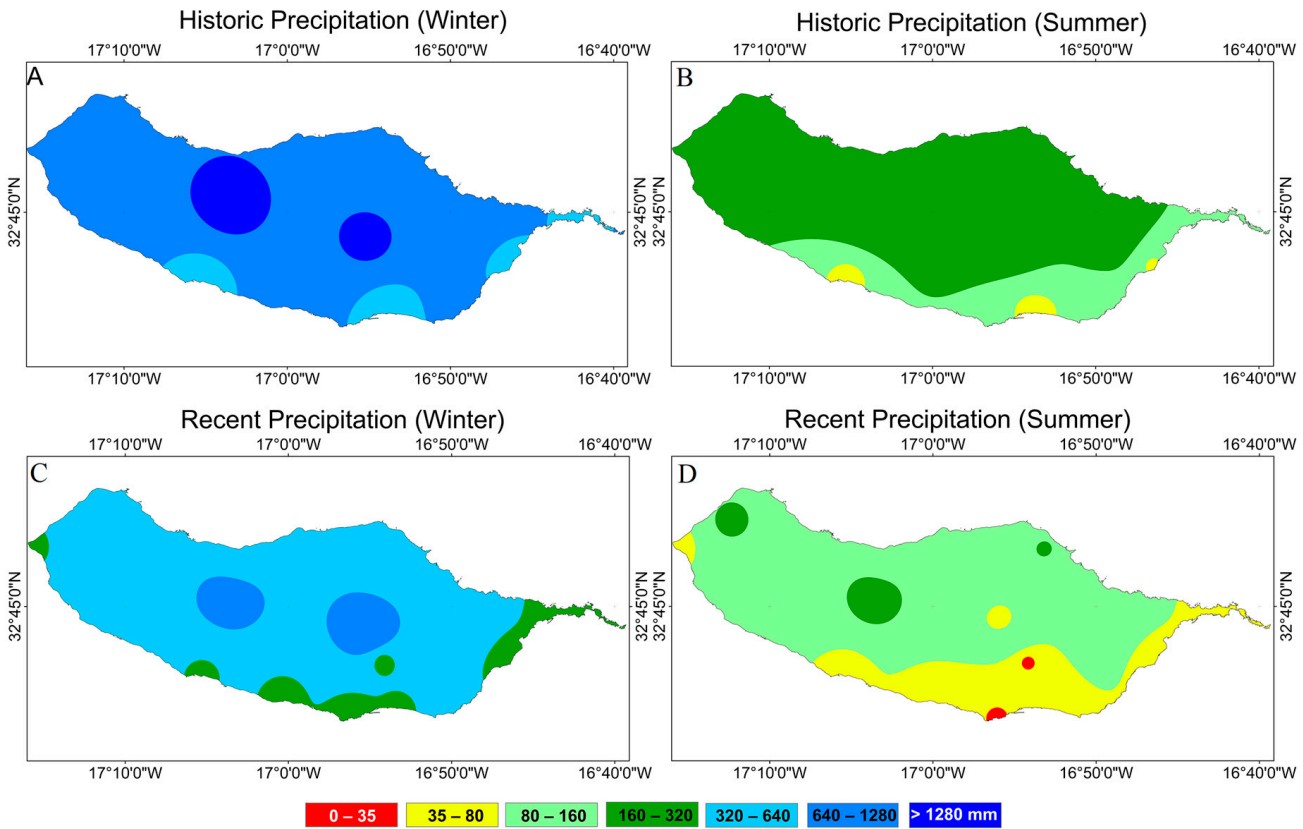

**Figure 3.** Interpolation of the accumulated precipitation of Madeira Island in extreme events of summer and winter, using the historical series (1960–1991), maps (**A**,**C**), and recent series (2010–2020) maps (**B**,**D**).

### 3.1.3. Comparison of Actual Climate Conditions with Future Climate Scenarios

The dramatic variation in average temperatures and accumulated precipitation is shown by spatialization and interpolation models that determine a need to understand if the proposed ΔA2 and ΔB2 scenarios [5] will be anticipated. These two scenarios will be characterized by an average annual increase in temperatures of 1.6 °C and 2.5 and a decrease of accumulated precipitation of 12.4 and 24.5%, respectively, which were foreseen in the QV area (Table 2).

**Table 2.** Monthly temperature increase and precipitation decrease variations in the two simulated scenarios, according to Gomes et al. [5].

| Month | Temperature Increases in ΔA2 (°C) | Temperature Increases in ΔB2 (°C) | Precipitation Decreases in ΔA2 (%) | Precipitation Decreases in ΔB2 (%) |
|---|---|---|---|---|
| January | 2.4 | 1.5 | −34 | −40 |
| February | 2.8 | 1.8 | −0.5 | −34 |
| March | 2.5 | 1.6 | −33 | −32 |
| April | 2.7 | 1.6 | −39 | −30 |
| May | 2.8 | 1.7 | −61 | −48 |
| June | 2.7 | 1.8 | 9 | 23 |
| July | 2.6 | 1.7 | 92 | 33 |
| August | 2.3 | 1.5 | 94 | −34 |
| September | 2.3 | 1.6 | −33 | −37 |
| October | 2.3 | 1.4 | −56 | −25 |
| November | 2.5 | 1.5 | −53 | −40 |
| December | 2.3 | 1.2 | −34 | −30 |

The scenarios ΔA2 and ΔB2 were modulated, using the historical series of temperature and precipitation data (1961–1990), applying the expected influence of anthropogenic activity and the increase in greenhouse gas emissions to estimate the expected climate changes [5]. Temperature and precipitation values for expected scenarios were compared with those of the historical meteorological series to obtain their monthly variation (Table 2). The meteorological data of the recent series obtained from the nearest QV station were used to compare the monthly average temperature and accumulated precipitation in the 2010 to 2020 period. The purpose of this simulation was to verify whether what theoretically would occur in the QV in 2070–2099, according to ΔA2 and ΔB2 scenarios, was or was not already anticipated. The actual climate conditions in QV can be visually and numerically verified in Figure 4A–D.

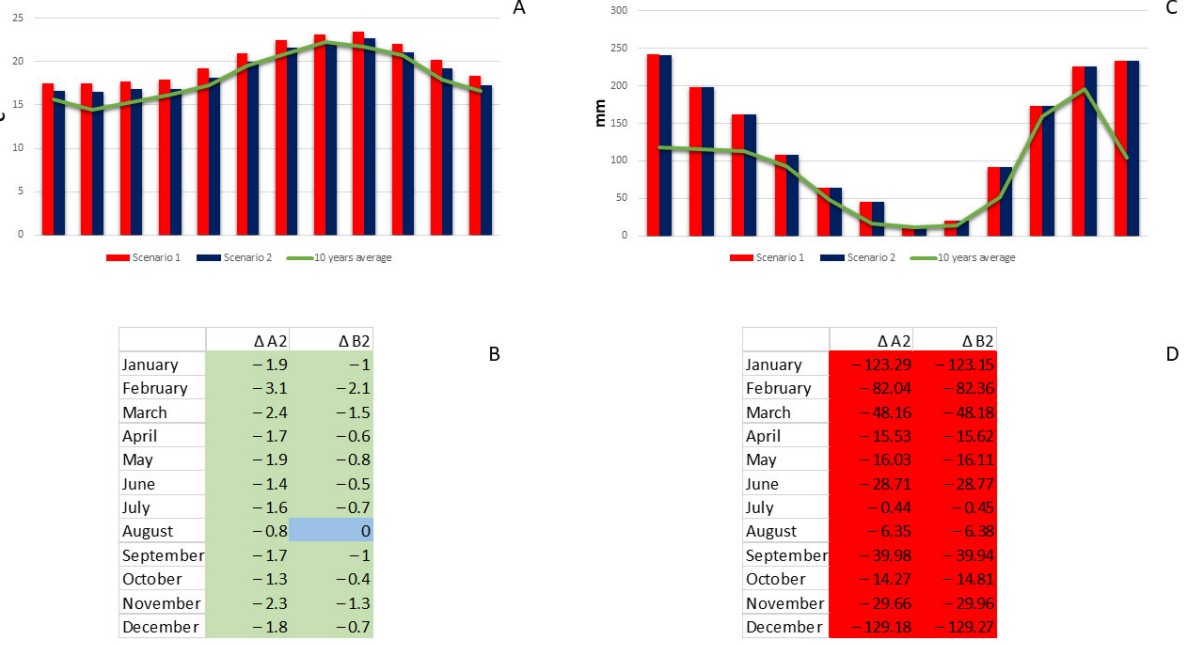

**Figure 4.** Comparison of recent series average (**A**) temperature and (**C**) accumulated precipitation (green line) with simulated ones for 2070–2099 scenarios ΔA2 red and ΔB2 blue columns. Difference between simulated average temperature and accumulated precipitation for the two scenarios (ΔA2 and ΔB2), with recent series of temperature and accumulated precipitation (Figure 4B and Figure 4D respectively).

Figure 4A shows the recent series (lines) and scenarios (ΔA2 red and ΔB2 blue columns) monthly average temperatures. In general, the average temperatures that would occur in 2070–2099 [5] have not been reached yet, even in the scenario ΔB2, where it is possible to observe a less dramatic increment in temperature (Figure 2B). It was found that the actual average temperatures are below the forecasted scenario for most months when compared with the recent series simulation. However, for the ΔB2, especially in summer extreme events (Figure 4B), the expected average temperature in QV has already or is close to being reached. Therefore, it is possible to anticipate that the average temperature is rising fast and worse than expected (see above), which is consistent with the last evaluation of climate changes made by the IPCC [26]. In the same way, the comparison of current accumulated precipitation (line) with the calculated ones for both scenarios (columns) (Figure 4B,D) showed an all-around year, aggravating the diminishment of the monthly precipitation. This diminishment affects both the winter and summer extreme events analyzed by the climate strategy [5].

Figure 4C,D quantifies the deviation between the average accumulated precipitation of the observed recent climate series (lines) and both climate scenarios (columns). Throughout the year, the studied agrosystem had well-below precipitation and the rainfed does not fulfill vineyards' crop water requirements, ranging between 400 and 1200 mm during its annual cycle, which manly condensate in a 6- to 7-month period between the stages of bud growth until the ripening [27].

### 3.2. Climate Water Balance

One of the major climate change constraints is imposed by water scarcity and drought, which can limit the agrosystem crop productivity and resiliency. Analyzing early data, it is possible to verify that, practically all year round, the QV already suffers a water shortage higher than expected from the extrapolations of climate conditions of ΔA2 and ΔB2 scenarios.

Although, to understand the influence of water shortage on agrosystem or crop productivity, the monthly field hydric balance, soil water capacity, and evapotranspiration potential for QV were calculated. Figure 5 shows the evolution of monthly hydric balance for QV, which is characterized by average precipitation of 53.2 mm·month$^{-1}$, totalizing accumulated precipitation of 639 mm per year and condensate in six months, between October and March. The figure shows that soil water surplus reaching 40 mm is registered between January and March, at the end of the winter extreme event. In the remaining-year period, a soil water deficit was recorded, with a maximum of −131.1 mm, starting in April and ending in October, covering all summer extreme events. During the soil water deficit period, crops and other living beings suffer drought stress and need to develop resilience strategies.

However, analyzing the vineyards growing in QV under such conditions, we hypothesize that some adaptation to the precipitation shortage could exist. Water shortage affects a major part of the vineyard production cycle, limiting nutrient uptake and transportation, which conditionate crop photosynthesis and yield. However, the influence of water shortage and soil hydric deficit on crops depends on the balance between soil hydric balance, evapotranspiration potential (ETP), and real evapotranspiration (ETR) and its transpiration and evaporation components. The annual ETP of QV, calculated by the Thornthwaite and Mather method, is 989.81 mm per year, ranging between 135.12 (August) and 42.24 mm (February). The ETP was higher than accumulated precipitation at 350.81 mm. The annual ETR in the QV was 575.5 mm. The monthly average ETP and ETR was 82.84 mm and 48.00 mm, respectively, showing a difference of 34.84 mm; both do not take into consideration the effect of irrigation. The comparison of variation in ETP and ETR and monthly soil water balance shows that the water atmospheric demand (625.99 mm) was higher than water available in the soil, e.g., accumulate precipitation (174.80 mm), during the soil water deficit (from April to October), giving an important water deficit of −451.19 mm. Therefore, to fulfill this gap and vegetation water demands, namely from vines, the agrosystem, e.g.,

soil, depends on field irrigation, the needs of which will increase with the be exacerbated by the ongoing decrease in accumulated precipitation and rise in average temperature. During the period of the study, 900 mm of water was used to overcome the water deficit and irrigate the vineyards until the grapes were repining, using, on average, a 150 mm month watering regime applied in three periodical irrigations.

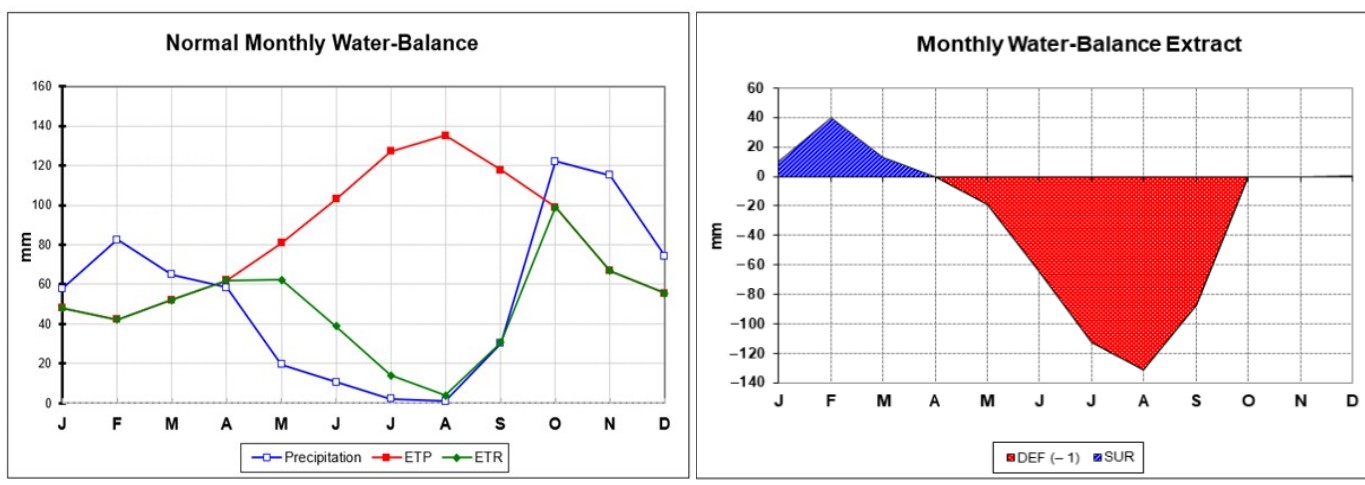

**Figure 5.** Calculation of water balance in the agrosystem of QV, including on the left monthly accumulate precipitation (blue line), Evapotranspiration potential (ETP, red line), and Real Evapotranspiration (ETR), and on the right monthly field hydric balance (blue diagram area) and soil water capacity (red diagram area).

*3.3. Soil Physicochemical Features or Properties*

*Calheta* county biophysical agrodiversity is characterized by five pedological complexes, which include eight different pedological units, with agricultural use [28]. The QV soils are chromic cambisol type, with *solum* mainly >50 cm, with some <50 cm, with some stoniness, that is, a little stoniness on the surface and some rocky outcrops. Organic matter (OM) in the soil reaches an average value of 3.1%, below the average OM value (5.9%), characteristic of Madeiran agricultural soils, according to data from the ISOPlexis Information and Documentation System (IDS) [29]. Physicochemical analysis indicates that the soil has a clayey or silty-clay texture, 49.6% field capacity, a cation exchange capacity (CEC) of 38.0, a saturation degree of 47.3, and an acidic pH (5.04–5.38 KCl method). The values of assimilable macronutrients range from 117.0 to 136.0 ppm for assimilable P, 296.0 and 640.0 ppm for assimilable K, 7.4 and 37.1 ppm for nitrates, and 2.0 and 5.9 ppm for nitrogen in the form of ammonia.

*3.4. Soil Biological Properties*

3.4.1. Microbiological Indicators

The influence of climatic conditions on QV soil diversity was monitored based on the average quantifications of the four groups of microorganisms listed in Table 3. Total bacteria and fungi and nitrogen-fixing bacteria showed values between $10^6$ and $10^7$ CFU/g (colony forming units) in dry soil. Denitrifying bacteria showed values around $10^3$ MPN/g (most probable number) of dry soil. The lowest values obtained for heterotrophic bacteria and fungi were observed in summer in both years. For nitrogen-fixing bacteria and denitrifying bacteria, a pattern along seasons could not be found.

An ANOVA analysis showed significant differences between seasons for total bacteria ($p = 0.01$) and total fungi ($p = 0.03$). Quantification obtained for bacteria in summer is significantly different from spring ($p = 0.02$) and autumn ($p = 0.02$), whilst values obtained for fungi in summer are significantly different from winter ($p = 0.046$) and spring ($p = 0.02$). Boxplots in Figure 6 show the variation in Log CFU per gram of soil obtained in samples for the different seasons.

**Table 3.** Average quantification of microorganism's colonies among seasons in *Quinta das Vinhas*, total bacteria and fungi, denitrifying, and nitrogen-fixing bacteria, for 2 years.

| Year | Season | Bacteria (CFU/g) | Fungi (CFU/g) | Nitrogen-Fixing Bacteria (CFU/g) | Denitrifying Bacteria (MPN/g) |
|---|---|---|---|---|---|
| 2018 | Winter | $1.19 \times 10^7$ | $1.60 \times 10^7$ | $1.35 \times 10^7$ | $5.09 \times 10^3$ |
| | Spring | $1.31 \times 10^7$ | $1.18 \times 10^7$ | $1.31 \times 10^7$ | $4.06 \times 10^3$ |
| | Summer | $7.60 \times 10^6$ | $9.22 \times 10^6$ | $1.38 \times 10^7$ | $5.62 \times 10^3$ |
| | Autumn | $1.74 \times 10^7$ | $1.29 \times 10^7$ | $2.35 \times 10^7$ | $1.27 \times 10^3$ |
| 2019 | Winter | $1.37 \times 10^7$ | $1.19 \times 10^7$ | $1.97 \times 10^7$ | $4.23 \times 10^3$ |
| | Spring | $2.77 \times 10^7$ | $2.31 \times 10^7$ | $2.61 \times 10^7$ | $1.08 \times 10^3$ |
| | Summer | $1.19 \times 10^7$ | $7.77 \times 10^6$ | $1.03 \times 10^7$ | $5.34 \times 10^3$ |
| | Autumn | $1.30 \times 10^7$ | $1.07 \times 10^7$ | $1.13 \times 10^7$ | $4.33 \times 10^3$ |

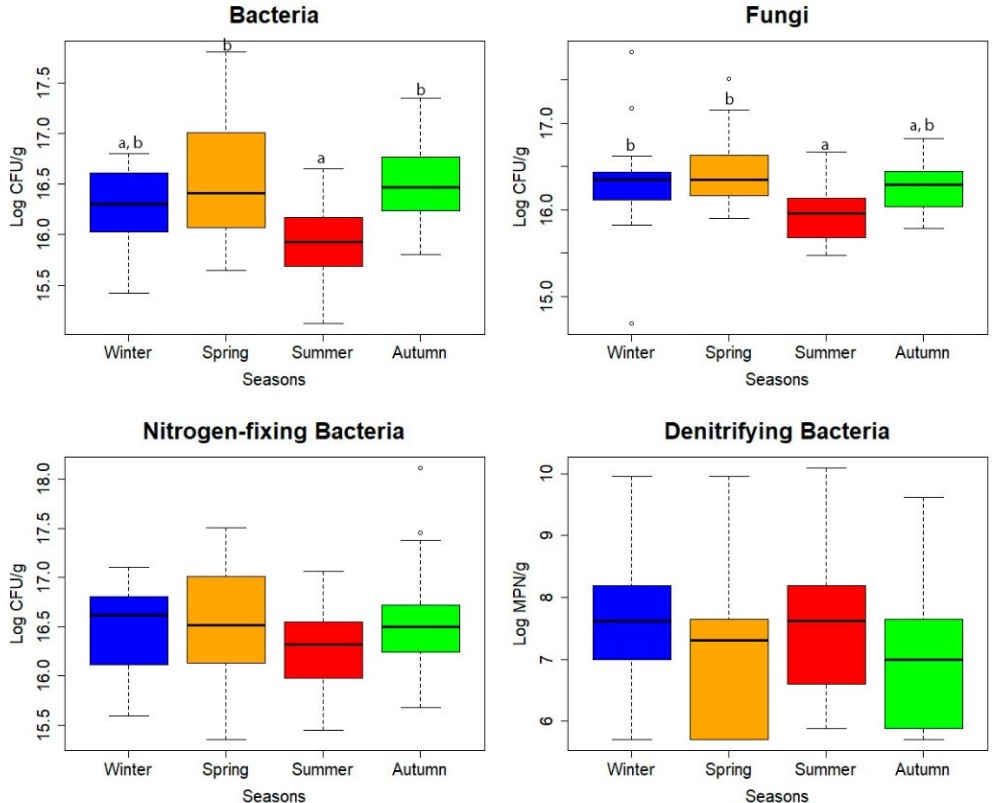

**Figure 6.** Variation in total bacteria and fungi and nitrogen-fixing and denitrifying bacteria abundance according to seasons. Boxes with different letters indicate a significant difference among them ($p \leq 0.05$). The letter a corresponds to the lowest values.

Microorganisms are important indicators of soil quality and, if monitored for long periods, it is possible to identify trends [30]. Microorganisms, in addition to being an important fraction of OM, are key players in carbon, nitrogen, and nutrient cycling [31]. In addition, they can play a role in the immobilization or transference of elements with the greenhouse gas effect [32]. Although two years of monitoring microbial indicators are not enough to ensure a trend, the variation in assessed indicators shows some patterns.

The number of viable bacteria and fungi per gram of soil obtained agrees with the expected range of $10^6$ to $10^9$ CFU per gram of soil for bacteria [33] and $10^3$ to $10^6$ CFU per gram of soil for fungi [34,35]. In addition, these values are similar to the average values found in other agricultural soils from the Madeira archipelago [29]. Bacteria and fungi show a similar pattern, having the highest values in intermediate seasons (autumn and spring) and the lowest in summer. The lowest values in summer were observed in both years. Summer was reported as the season for optimal growth and a higher abundance

of microorganisms, mainly bacteria [36–39]. However, those studies were carried out in regions with higher precipitations during summer, which does not occur in this study case, where severe drought conditions are observed (Figures 3 and 5). The observed climate conditions in QV contributed to the change in soil physicochemical properties that may lead to a less suitable environment for microbial growth [39]. In our study, correlations between precipitation and assimilable K (r = −0.799, $p < 0.01$), nitrates (r = 0.732, $p < 0.01$), ammonial nitrogen (r = 0.769, $p < 0.01$), manganese (r = 0.624, $p \leq 0.05$), and iron (r = 0.611, $p < 0.01$) were observed, giving support to this hypothesis.

Nitrogen is a key nutrient for plants; however, under specific conditions, its release into the atmosphere can contribute to global warming. Nitrogen-fixing bacteria are capable to fix atmospheric $N_2$ into ammonium, which can be taken up by plant roots [40]. On the other hand, denitrifying bacteria can reduce nitrate to gaseous nitrogen compounds (NO, $N_2O$, and $N_2$), being responsible for the return of part of the nitrogen into the atmosphere [41]. These two processes are, therefore, important to maintain the overall nitrogen balance in the soil. Nevertheless, in agriculture, denitrification means the loss of nitrogen, which can result in a decrease in the productivity and quality of harvested products [42]. Only 1% or less of the bacteria that carry these processes can be detected in the growth media, so its monitoring only provides us a reference value in situations of imbalance [40,41,43]. The values for putative nitrogen-fixing bacteria range between $10^3$ and $10^8$ CFU per gram of soil [40,44,45] and between $10^3$ and $10^7$ viable cells per gram of soil for putative denitrifying bacteria [40,41,43]. The observed values for both indicators in QV are within the range of bibliographic data. Further, the number of putative nitrogen-fixing bacteria is higher than putative denitrifying bacteria (Table 3). Regarding variation among seasons, there are no significant differences within these groups (Figure 6). However, the average proportion of viable cells between nitrogen-fixing and denitrifying bacteria in QV is 4149 to 1, respectively. These values range between 24,103 (spring, 2018) and 1929 (summer, 2019) to one denitrifying bacteria. In both monitoring years, summer was the season that was farther below the usual average ratio between nitrogen-fixing/denitrifying bacteria and that could be determined. Braker and colleagues demonstrated that the increase in temperature leads not only to an increase in denitrifying bacteria abundance but also to an increase in denitrification activity [46]. Long periods of warming followed by drought seem to have the same effect, perhaps due to a decrease in $O_2$ concentration in the soil [47]. The observed imbalance between N-fixation and N-reduction, during summer, in QV can be worrying not only for the loss of nitrogen, but also because it contributes to the emission of greenhouse gases.

3.4.2. Microbiological Diversity Indices and Seasonal Variation

The soil microbiological communities in QV have been analyzed using molecular markers to assess the potential impact of climate on microorganism diversity. The T-RFLP method appeared as a rapid, robust, and high-throughput molecular tool for assessing microbial community structure. Diversity indices are commonly used in ecology and provide information about the community but can be applied to assess the microbiological community. Microbial diversity was evaluated through species richness (S), abundance (N), Shannon–Wiener (H′), Corrected Evenness (E′), and Simpson (D) indices and are presented in Figure 7 and Table 4.

The richness of the four groups under study was calculated based on the number of different terminal restriction fragments (T-RFs), representing taxonomic units, and the abundance was based on their peak area, after determining "true peaks" and eliminating background noise with T-REX software [18]. Shannon–Wiener diversity index, which is calculated by the number and weight of T-RFs in the community, Corrected Evenness, which can be interpreted as community homogeneity and Simpson index, which measures the probability that two individuals selected at random from a sample belong to the same species, were calculated according to the number and abundance of T-RFs.

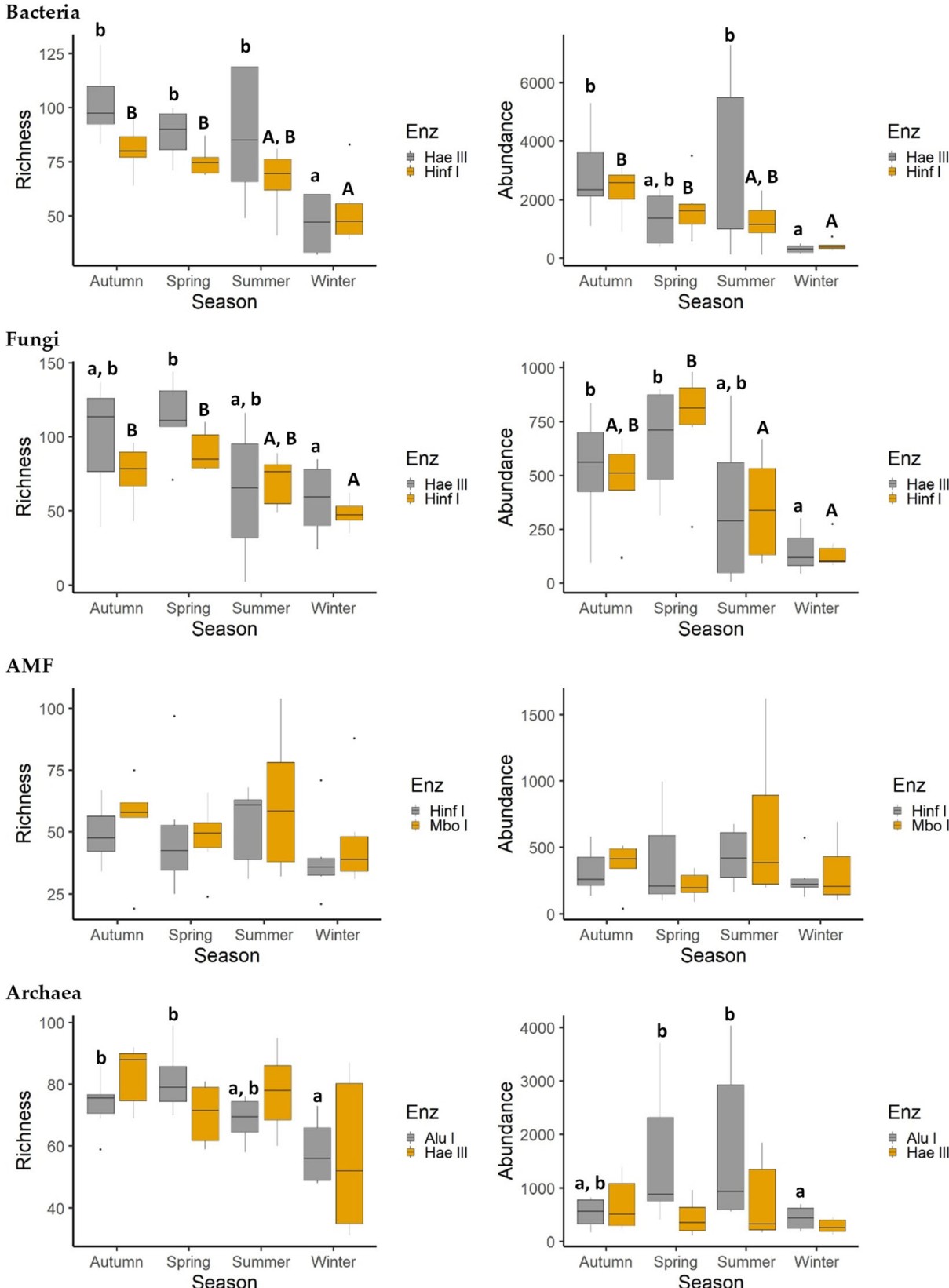

**Figure 7.** Variation in richness and abundance of Bacteria, Fungi, AMF, and Archaea, according to seasons. Boxes with different letters indicate a significant difference among them ($p \leq 0.05$). The letter a corresponds to the lowest values. Lowercase letters are for grey boxes and uppercase letters are for orange boxes.

**Table 4.** Shannon–Winner diversity (H′), Corrected Evenness (E′), and Simpson index (D) of the four microorganisms' groups, according to seasons.

| | | H′ | | E′ | | D | |
|---|---|---|---|---|---|---|---|
| | Season | Enz 1 | Enz 2 | Enz 1 | Enz 2 | Enz 1 | Enz 2 |
| Bacteria | Winter | 4.56 ± 0.64 a | 4.61 ± 0.54 | 0.77 ± 0.07 | 0.76 ± 0.05 | 0.07 ± 0.03 b | 0.06 ± 0.02 |
| | Spring | 5.63 ± 0.27 b | 4.97 ± 0.10 | 0.84 ± 0.04 | 0.77 ± 0.02 | 0.03 ± 0.01 a | 0.04 ± 0.00 |
| | Summer | 5.39 ± 0.38 a,b | 4.72 ± 0.09 | 0.81 ± 0.03 | 0.74 ± 0.02 | 0.03 ± 0.01 a,b | 0.06 ± 0.01 |
| | Autumn | 5.58 ± 0.27 b | 4.90 ± 0.11 | 0.82 ± 0.03 | 0.76 ± 0.02 | 0.02 ± 0.01 a | 0.05 ± 0.01 |
| Fungi | Winter | 4.69 ± 0.92 | 4.66 ± 0.48 a | 0.47 ± 0.29 a | 0.62 ± 0.17 a | 0.07 ± 0.04 | 0.06 ± 0.02 b |
| | Spring | 5.81 ± 0.33 | 5.50 ± 0.26 b | 0.79 ± 0.06 a,b | 0.80 ± 0.04 b | 0.03 ± 0.01 | 0.03 ± 0.01 a |
| | Summer | 4.56 ± 1.94 | 5.29 ± 0.09 a,b | 0.60 ± 0.26 a,b | 0.74 ± 0.10 a,b | 0.13 ± 0.23 | 0.04 ± 0.01 a,b |
| | Autumn | 5.65 ± 0.42 | 5.36 ± 0.23 b | 0.81 ± 0.06 b | 0.81 ± 0.03 b | 0.03 ± 0.01 | 0.04 ± 0.01 a,b |
| AMF | Winter | 3.55 ± 1.2 | 3.56 ± 0.9 | 0.55 ± 0.24 | 0.48 ± 0.20 | 0.2 ± 0.15 | 0.20 ± 0.11 |
| | Spring | 3.97 ± 0.93 | 3.88 ± 0.25 | 0.58 ± 0.17 | 0.52 ± 0.15 | 0.14 ± 0.08 | 0.12 ± 0.02 |
| | Summer | 3.35 ± 0.67 | 3.85 ± 0.63 | 0.48 ± 0.13 | 0.58 ± 0.12 | 0.22 ± 0.09 | 0.14 ± 0.05 |
| | Autumn | 4.08 ± 0.54 | 3.95 ± 0.46 | 0.61 ± 0.12 | 0.61 ± 0.15 | 0.13 ± 0.04 | 0.13 ± 0.06 |
| Archaea | Winter | 4.39 ± 0.50 | 4.87 ± 1.14 | 0.77 ± 0.16 | 0.67 ± 0.10 | 0.09 ± 0.05 | 0.07 ± 0.05 |
| | Spring | 4.94 ± 0.56 | 5.29 ± 0.57 | 0.80 ± 0.07 | 0.75 ± 0.11 | 0.06 ± 0.04 | 0.04 ± 0.02 |
| | Summer | 4.34 ± 0.63 | 5.34 ± 0.52 | 0.81 ± 0.10 | 0.66 ± 0.12 | 0.11 ± 0.07 | 0.04 ± 0.03 |
| | Autumn | 5.15 ± 0.72 | 5.35 ± 0.81 | 0.80 ± 0.13 | 0.77 ± 0.10 | 0.05 ± 0.03 | 0.05 ± 0.04 |

**Note**: For bacteria and fungi, Enz 1: *Hae III*/Enz 2: *Hinf I*; for AMF, Enz 1: *Hinf I*/Enz 2: *Mbo I*; for Archaea, Enz 1: *Hae III*/Enz 2: *Alu I*. Different letters indicate a significant difference among them ($p \leq 0.05$). The letter a corresponds to the lowest values.

Fungi and bacteria showed the same pattern as in classical microbiology for richness and abundance (Figure 7). Autumn and spring were the seasons with the highest values for these indices and summer and winter with the lowest. However, only winter showed significant differences from autumn and spring for T-RFs generated by both enzymes ($p \leq 0.05$) in both microorganism groups.

For bacteria communities, richness and abundance are higher in autumn and decrease in spring, then summer and winter. Enzyme *Hae III* was able to originate, in general, more T-RFs in the bacteria community than the enzyme *Hinf I* and made it possible to also find a significant difference between winter and summer ($p \leq 0.05$) regarding richness and abundance. Winter environmental conditions proved to be less suitable for the communities of bacteria present in QV when compared to other seasons. This is also evident from the results of Shannon–Wiener diversity and Simpson index (Table 4). The lower the value of H′, the smaller the diversity and, for the D index, the higher the value, the less even and rich the community. It is also possible to observe that there was no significant difference between winter and spring when it comes to abundance. This means that environmental conditions in spring favored a high number of taxonomic units, but the total size of the community was like winter. However, it does not seem to affect negatively the diversity in the community. Different from bacteria, fungal communities are richer and more abundant in spring (Figure 7). In this case, the enzyme *Hinf I* was more successful in differentiating the diversity in fungal communities among seasons (Table 4). Fungal communities from winter and summer are less diverse, although only winter is significantly different from spring and autumn. Additionally, the evenness index shows autumn as the season where the community is more homogeneous and winter less, the latter with values around 0.5.

An early study in vineyard soils contemplating only the two extreme seasons (winter and summer) could not identify significant differences in bacterial and fungal community structure [48]. They found a core microbiome that was conserved in both seasons, varying

only in quantity. Physicochemical parameters were indicated as the main drivers of communities' structure. In our study, we also compared these seasons with intermediate seasons and the negative impact of the extreme seasons on soil bacterial and fungal communities is evident, which might affect soil functioning.

Arbuscular mycorrhizal fungi (AMF) did not show significant differences for the analyzed indices, meaning that this fungal group is quite stable during seasons. However, the number of T-RFs originated by enzyme *Mbo I* (richness) show a slight decrease from autumn to spring and then summer and winter (Figure 7). It is also possible to see that Shannon–Wiener diversity is higher in autumn and spring and lower in summer and winter (Table 4). For evenness, the values were lower than the ones found for total fungi, which means that in this group, there were some taxonomical units in higher numbers than others and this is especially true in summer. Simpson indices confirm this because the values are higher than values found in other groups, so some species are dominant. The absence of significant seasonal changes in this community may be due to a predominance of the genus *Glomus*, which is, in general, more stable between extreme seasons and different soil conditions [49]. On the other hand, the absence of significant differences in these indices among seasons does not necessarily mean that there is no change within the community. It is possible that some species or genera are being replaced by others over the seasons [49,50].

Finally, archaea showed significant differences only for T-RFs generated by the enzyme *Alu I* (Figure 7). For richness, significant differences between winter and autumn ($p \leq 0.05$) and spring ($p < 0.01$) were found again, but regarding abundance, the differences were between winter and spring ($p \leq 0.05$) and summer ($p \leq 0.05$). No significant differences were found for H', D, and E (Table 4). This means that the variation in environmental conditions among seasons had no significant impact on the diversity of archaea in the QV communities, although autumn seems to have better conditions for this group, as it was the season where the group was more diverse and even.

### 3.4.3. Relationship between Microbiological Diversity Indices, Climatic Data, and Edaphic Parameters

The influence of local climatic conditions on the microbiological indicators of the agrosystem was assessed through the identification of correlations between diversity indices of the four microorganisms' groups and climatic data. For the bacteria group, negative correlations were observed between the evenness index and minimum and average temperatures (r = −0.719 and r = −0.731 $p \leq 0.05$, respectively) and with relative humidity (r = −0.710 $p \leq 0.05$) for *Hinf I* enzyme. Related to AMF, for enzyme *Hinf I*, a negative correlation was observed between abundance and precipitation (r = −0.744 $p \leq 0.05$). Archaea show a positive correlation between abundance and minimum temperature (r = 0.841 $p < 0.01$) for *Hae III*. Fungi showed no significant differences between diversity indices and climatic data. To summarize, the temperature impacted the bacteria evenness and archaea abundance and the precipitation negatively impacted the AMF abundance, but no direct influence of other climatic parameters on microbiological groups was observed.

The existence of correlations between the four microorganisms' groups' diversity indices and edaphic parameters was also assessed. Bacteria diversity indices based on *Hae III* showed several significant correlations with edaphic parameters. Bacteria abundance was negatively correlated with nitrates and ammoniacal nitrogen (r = −0.988 $p \leq 0.05$, r = −0.996 $p \leq 0.05$, respectively). Finally, the bacteria richness and Shannon diversity were positively correlated with CEC (r = 0.999 $p < 0.01$, r = 0.959 $p \leq 0.05$, respectively). At the same time, bacteria diversity indices based on *Hinf I* showed the following significant correlations with edaphic parameters, assimilable K was negatively correlated with Simpson index (r = −0.954 $p \leq 0.05$) and positively with abundance (r = 0.991 $p < 0.01$) and Shannon index (r = 0.955 $p \leq 0.05$). Finally, evenness was positively correlated with pH (r = 0.994 $p < 0.01$).

Regarding archaea, according to the enzymes *Alu I* and *Hae III* restriction products, the micronutrients (Cu) showed a negative correlation with richness and abundance (r = −1.00

$p \leq 0.01$). A negative correlation between Shannon diversity and assimilable P (r = −0.979 $p \leq 0.05$) and between evenness and assimilable P (r = −0.979 $p \leq 0.05$) and a positive correlation between richness and assimilable K (r = 0.970 $p \leq 0.05$) and between Simpson index and organic matter (r = 0.955 $p \leq 0.05$) were also observed for T-RFs originated by *Alu I*. At the same time, the Simpson index was negatively correlated with assimilable K (r = −0.972 $p \leq 0.05$), evenness and Shannon diversity were positively correlated with assimilable K (r = 0.972 $p \leq 0.05$, r = 0.962 $p \leq 0.05$, respectively), and richness with CEC (r = 0.971 $p \leq 0.05$), according to enzyme *Hae III*.

Summarizing pH has a positive influence on bacteria and archaea evenness and bacteria richness. The CEC has a positive influence on bacteria richness and diversity and archaea richness. The organic matter positively influences the Simpson index for archaea. The changes in macronutrients, e.g., nitrates and ammoniac nitrogen, have a negative impact on bacteria abundance. At the same time, assimilable K positively impacts bacteria abundance and Shannon diversity, archaea richness, and Shannon diversity and impacts negatively on bacteria and archaea Simpson index. The assimilable P negatively influences archaea Simpson index. The micronutrients (Cu) have a negative impact on archaea richness and abundance.

Regarding the influence of edaphic parameters on fungi, community correlations were observed with assimilable K, CEC, and micronutrients (Cu). Simpson index was negatively correlated with assimilable K but positively with assimilable P for enzyme *Hae III* ($p \leq 0.01$). Shannon diversity was positively correlated with assimilable K and negatively with assimilable P, for both enzymes ($p \leq 0.01$). Richness was also positively correlated with assimilable K (r = 0.981 $p \leq 0.05$) for *Hae III*. CEC was positively correlated with the richness of T-RFs originated by *Hinf I* (r = 0.958 $p \leq 0.05$) and with abundance (r = 0.999 $p < 0.01$) of *Hae III*. Changes in Cu negatively influenced the richness of products generated by *Hinf I* and the abundance of products generated by *Hae III* ($p \leq 0.01$).

A negative correlation was observed in AMF, for T-RFs originated by enzyme *Mbo I*, between the Shannon diversity and soil moisture (r = −0.747 $p \leq 0.05$), evenness and assimilable P (r = −0.960 $p \leq 0.05$), Simpson index and assimilable K (r = −0.996 $p < 0.01$), richness and nitrates (r = −0.960 $p \leq 0.05$). Positive correlations were observed between Simpson index and soil moisture (r = 0.785 $p \leq 0.05$), Shannon diversity and assimilable K (r = 0.968 $p \leq 0.05$), and evenness and assimilable K (r = 0.958 $p \leq 0.05$). A negative correlation, regarding enzyme *Hinf I*, between Simpson index and pH (r = −0.970 $p \leq 0.05$), was observed. Abundance, Shannon diversity, and evenness were positively correlated with pH (r = 0.964 $p \leq 0.05$, r = 0.977 $p \leq 0.05$, r = 0.970 $p \leq 0.05$, respectively).

In short, soil moisture positively affects the AMF Simpson index and negatively the Shannon diversity. The pH positively influences AMF abundance, evenness, and Shannon diversity and negatively AMF Simpson index. CEC has a positive correlation with fungi richness and abundance. The changes in macronutrients, e.g., assimilable K, positively impact fungi and AMF Shannon diversity, fungi richness and evenness, and negatively fungi and AMF Simpson index. On the other hand, assimilable P and the micronutrients (Cu) negatively impacted the fungal community. Finally, assimilable P and nitrates negatively affect AMF evenness and richness, respectively.

These results demonstrate a direct influence of some climatic conditions on microbial communities, especially for bacteria, archaea, and the fungal functional group AMF. However, the diversity and abundance of microbial communities are more closely associated with soil physicochemical parameters. These factors seem to contribute to the fluctuations observed across seasons. Temporal variation in climate conditions can have direct or indirect effects on soil properties, such as CEC, pH, and nutrient availability [51,52], and on plant diversity; these factors, in turn, can have an impact on soil microbial communities [53]. For example, potassium is correlated negatively with precipitation and soil moisture in QV ($p \leq 0.01$); one of the reasons could be nutrient leaching [54,55]. Our results showed that low concentrations of this nutrient lead to a negative impact on soil microbial communities, which may explain, in part, the lower abundance and diversity in winter. On the other

hand, the low abundance and diversity of soil microbial communities in summer may be more related to higher temperatures and lower water availability.

### 3.5. Floristic Survey and Plant Diversity Indicators

The spontaneous species occurrence and diversity were monitored in QV to assess the possible influence of climatic conditions on plant communities as indicators. The results of the plant surveys as an average number of species and specimens' quantifications are listed in Table 5. Floristic composition data were gathered per trimester, along transepts, conducting a census in six areas of the QV agrosystem.

**Table 5.** Distribution of total species and individuals, as well as the ranking of the three more representative life forms in the QV agrosystem. Species were also grouped according to Raunkiaer's Life Form classification [56].

| Year | Season | Total Species | Total Individuals | Therophyte (Th) | Hemicryptophyte (H) | Geophyte (G) | Remaining Classes (R) |
|------|--------|---------------|-------------------|-----------------|---------------------|--------------|-----------------------|
| 2018 | Spring | 53 | 605 | 42 | 7 | 3 | 1 |
| | Summer | 20 | 27 | 11 | 4 | 2 | 3 |
| | Autumn | 48 | 425 | 33 | 6 | 7 | 2 |
| | Winter | 68 | 713 | 44 | 11 | 7 | 6 |
| 2019 | Spring | 43 | 164 | 22 | 10 | 6 | 5 |
| | Summer | 56 | 289 | 29 | 13 | 6 | 8 |
| | Autumn | 66 | 1004 | 36 | 10 | 9 | 11 |
| | Winter | 55 | 238 | 35 | 9 | 5 | 6 |

A total of 409 species (with repetitions) occurred during the monitoring period and the project's timeline. Two values stand out; winter 2018 had the highest number of species present (68) and summer 2018, with the lowest number of species present (20).

In total, 3465 individuals were monitored in six sampling areas. The 2019 autumn and the summer of 2018 counts stand out with the highest (1004 individuals) and the lowest (27) number of individuals counted. Although there is seasonal variation in individuals in the agroecosystem, this variation in the number of individuals is highly related to the regular control of adventitious plants (mechanical weeding).

Surveyed plant diversity in the QV agrosystem was ranked by species, taxonomic family, ecological status, and life form classification (Table 6). In total, 107 species were inventoried. The three families with the highest number of species are the Compositae with 18 species, Poaceae with 14, and Leguminosae with 12 species. According to the ecological classification, in the agrosystem, there are 62 native or probable native species (57.9% of the total number of surveyed species), 44 are probable introduced or introduced species (41.1% of the total), and 1 endemism (0.9%). The only endemic species is the Macaronesian laurel (*Laurus novocanariensis*). These figures show the existence of significant plant diversity and species richness, which is a good indicator of the overall agrosystem condition.

The three most abundant life forms in this agrosystem are: Therophytes (Th) with a total of 252 species (with repetition) occurring during the monitoring time; Hemicryptophytes (H) with a total of 70 species (with repetition); and Geophytes (G) with a total of 45 species (with repetition). In the QV agrosystem, the 20 most frequent species, in terms of life form, are represented by therophytes with 65% (or 63.3% of total accounts with repetitions), hemicryptophytes with 20 (or 16.3% of total), and geophytes 15 (11.1% of total). These numbers of life form species occurrences give the percentage ratios of (Th), (H), and (G), which allows us to classify the agrosystem into extreme warmth and drought climate zones because of the prevalence of Th and disbalance of H life plant forms [56]. Therophytes include annual plants dispersed by seed at less-advantageous times. These species have

a greater number of species in the autumn and winter seasons, while in summer, their numbers decrease. The Hemicryptophytes are species with dormant buds on the upper soil surface and Geophytes species are characterized by bulbs, rhizomes, tubers, or roots that can function as dormant parts. The number of Hemicryptophytes and Geophytes varies little according to the seasons of the year, with an increase in species in the summer and autumn of 2019.

**Table 6.** Ranking of the QV agrosystem 20 most surveyed plant species and their respective taxonomic family, ecological status, and life form classification during the project's duration.

| Species | Family | Status | Life Form |
|---|---|---|---|
| *Bidens pilosa* L. | Compositae | Introduced | Therophyte |
| *Sonchus oleraceus* L. | Compositae | Native probable | Therophyte |
| *Conyza sumatrensis* (Retz.) E. Walker | Compositae | Introduced | Therophyte |
| *Convolvulus arvensis* L. | Convolvulaceae | Native | Geophyte |
| *Fumaria muralis* W.D.J.Koch | Papaveraceae | Native | Therophyte |
| *Malva parviflora* L. | Malvaceae | Native | Therophyte |
| *Solanum nigrum* L. | Solanaceae | Native probable | Therophyte |
| *Helminthotheca echioides* (L.) Holub | Compositae | Native probable | Therophyte |
| *Lactuca serriola* L. | Compositae | Intr. Probable | Hemicryptophyte |
| *Avena barbata* Link | Poaceae | Native probable | Therophyte |
| *Calendula arvensis* L. | Compositae | Native | Hemicryptophyte |
| *Stachys arvensis* L. | Lamiaceae | Native | Therophyte |
| *Allium neapolitanum* Cirillo | Amaryllidaceae | Intr. Probable | Geophyte |
| *Erodium moschatum* (L.) L'Hér. | Geraniaceae | Native | Therophyte |
| *Bromus catharticus* Vahl | Poaceae | Introduced | Hemicryptophyte |
| *Polycarpon tetraphyllum* L. | Caryophyllaceae | Native | Therophyte |
| *Medicago polymorpha* L. | Fabaceae | Native | Therophyte |
| *Ipomoea indica* (Burm.) Merr. | Convolvulaceae | Introduced | Geophyte |
| *Bituminaria bituminosa* (L.) C.H.Stirt. | Fabaceae | Native | Hemicryptophyte |
| *Senecio vulgaris* L. | Compositae | Native probable | Therophyte |

Correlations between the edaphoclimatic factors and life forms showed that Th has a strong negative correlation ($p < 0.01$) with average, maximum, and minimum temperature, something to be expected for annual plants, suspending its life cycle with the rising temperature. A less-strong negative correlation between relative humidity and radiation ($p \leq 0.05$) was also observed, which corroborates the early statement. For the H plant life form, there is a strong positive correlation ($p < 0.01$) between this indicator and soil pH, measured both on $H_2O$ and KCl suspensions. Finally, G showed a strong negative correlation ($p < 0.05$) with average and minimum temperature and strong positive correlation with soil pH measured in KCl and a less-strong correlation with soil pH measured in $H_2O$ ($p \leq 0.05$).

The results of the plant survey were used to calculate five biodiversity indices obtained to ascertain the status of wild vegetation (Table 7) and try to identify possible trends in climate influence on this indicator. The Species Richness (S) ranges between 9.33 ± 3.01, in Autumn 2019, and 1.80 ± 0.40, in summer 2018, the average number of species per square meter. Species abundance (N) ranged from 167.33 ± 72.07, in Autumn 2019, and 5.40 ± 3.50, in summer 2018. The highest value weighting dominant species, Shannon Index (H′), was obtained again in the autumn of 2019, with a value of 2.38 ± 0.78. The summer of 2018

had the lowest diversity (0.66 $\pm$ 0.38), being in line with all diversity indices. Corrected Evenness (E′) showed, throughout all seasons, homogenized values, except for the summer of 2018, with a value close to zero. The remaining results have relatively high homogeneity values, ranging between 0.49 $\pm$ 0.29, in Spring 2019 and 0.70 $\pm$ 0.19, in Autumn 2019. The data show that the agrosystem presents a native flora with relative homogeneity among seasons where there are no species that overwhelmingly dominate in the agrosystem. The Hill's Diversity index (N2) results are in line with the Shannon index (H′), with values ranging between 1.56 $\pm$ 0.39, in summer 2018, and 4.67 $\pm$ 2.45, in autumn 2019.

**Table 7.** Mean values with their respective standard deviation for Species richness (S), Species abundance (N) Shannon–Wiener diversity index (H′), Corrected Evenness (E′), and Hill's Index (N2) for the floristic composition of QV agrosystem.

| Year | Season | S | N | H′ | E′ | N2 |
|------|--------|---|---|----|----|----|
| 2018 | Winter | 6.17 $\pm$ 2.03 b | 118.83 $\pm$ 64.62 b | 1.62 $\pm$ 0.43 a,b | 0.56 $\pm$ 0.11 b | 2.51 $\pm$ 0.60 a,b |
|  | Spring | 8.33 $\pm$ 3.20 b | 100.83 $\pm$ 72.29 b | 2.13 $\pm$ 0.79 b | 0.58 $\pm$ 0.28 b | 3.96 $\pm$ 1.90 b |
|  | Summer | 1.80 $\pm$ 0.40 a | 5.40 $\pm$ 3.50 a | 0.66 $\pm$ 0.38 a | 0.00 $\pm$ 0.00 a | 1.56 $\pm$ 0.39 a |
|  | Autumn | 8.83 $\pm$ 2.11 b | 71.17 $\pm$ 28.61 a,b | 2.03 $\pm$ 0.47 b | 0.67 $\pm$ 0.08 b | 3.43 $\pm$ 0.94 a,b |
| 2019 | Winter | 5.83 $\pm$ 3.31 a,b | 39.67 $\pm$ 19.54 a | 1.54 $\pm$ 1.16 | 0.53 $\pm$ 0.30 | 2.98 $\pm$ 1.82 |
|  | Spring | 3.33 $\pm$ 1.51 a | 27.33 $\pm$ 12.04 a | 1.09 $\pm$ 0.70 | 0.49 $\pm$ 0.29 | 2.01 $\pm$ 0.94 |
|  | Summer | 4.67 $\pm$ 1.97 a | 48.17 $\pm$ 43.23 a | 1.51 $\pm$ 0.79 | 0.55 $\pm$ 0.21 | 2.74 $\pm$ 1.46 |
|  | Autumn | 9.33 $\pm$ 3.01 b | 167.33 $\pm$ 72.07 b | 2.38 $\pm$ 0.78 | 0.70 $\pm$ 0.19 | 4.67 $\pm$ 2.45 |

Different letters indicate a significant difference among them ($p \leq 0.05$). The letter a corresponds to the lowest values.

Analyzing the results for 2018, the summer season has, in general, significant differences for all the biodiversity measures used, compared to other seasons. In all indices for summer scored the lowest values (a), meaning that it is the most unfavorable season for general plant diversity in this agrosystem. The autumn season of 2019 shows significant differences from other seasons, but only in two biodiversity measures, S and N. However, in this case, autumn stands out by having the highest biodiversity indices out of the four seasons.

Analyzing, in detail, the year 2018, the S, summer, shows significant differences ($p \leq 0.05$) from all other seasons, with a low number of occurring species (1.80 $\pm$ 0.40). The N, summer, shows statistical differences ($p \leq 0.05$) with winter and spring, having the smallest individual count (5.40 $\pm$ 3.50). Summer H′ shows the lowest value (0.66 $\pm$ 0.38), having significant differences ($p \leq 0.05$) with spring and autumn. Winter does not show any statistical difference from all other seasons. In the E′ index, the results show that summer had the worst results (0.00 $\pm$ 0.00) on species homogeneity, having significant differences ($p \leq 0.05$) from all other seasons. Overall, except for summer, species had average homogeneity with autumn having the best result (0.67 $\pm$ 0.08). For N2 index, the results show that there is a significant difference ($p \leq 0.05$) in this Biodiversity Index between summer (1.56 $\pm$ 0.39) and spring (3.96 $\pm$ 1.90).

Analyzing, in detail, the year 2019, for S, autumn has significant differences ($p \leq 0.05$) with spring and summer, but not with winter. Autumn has a high number of species (9.33 $\pm$ 3.01) occurring compared to other seasons. The N, autumn shows statistical differences ($p \leq 0.05$) with all other seasons, having the highest individual count (167.33 $\pm$ 72.07). The remaining indices did not show statistically significant differences between the seasons.

A correlation between the edaphoclimatic factors and biodiversity indices showed that species richness (S) has a strong correlation ($p < 0.01$) with average and minimum temperature and soil pH and less-strong correlations with Relative Humidity and radiation ($p \leq 0.05$). Further, species abundance (N) shows a strong correlation ($p < 0.01$) with maximum temperature and soil humidity.

Therefore, the obtained results for plant diversity show a significant variation in species richness and abundance, and plant diversity in summer is associated with the prevalence of some plant life forms, e.a. therophytes, hemicryptophytes, and geophytes. However, the plant life forms [56] approach to link the plant diversity with climate and then use them as plant indicators to monitor ongoing climate changes is not always consensual [57]. The therophytes, hemicryptophytes, and geophytes plant life forms have been associated early with climate and temperature, precipitation, and insolation [58], which are major drivers of climate change. Therophytes and geophytes have in common the leaf–life form–root (LLR) strategy to cope the extreme climate conditions [59]. QV agrosystem is characterized by a prevalence of therophytes, which are often found in warm, severe drought and eroded zones, so it can be concluded that among these life forms, the suitable indicators of the negative shift in climate should be identified. Therefore, we hypothesize that they can be successfully used to monitor climate changes at an agrosystem scale.

### 3.6. Insect Survey and Diversity Indicators

Changes in the occurrence of insects in the agrosystem because of alterations in climate conditions are expected, with a drift to plagues increasing. The impact of the main drivers of climatic changes, such as higher atmospheric $CO_2$, higher temperatures, and a reduction in soil moisture, can affect the population dynamics of insect pests, with the increase in temperature favoring higher metabolic and development rates, reproduction, and survival capabilities [60,61]. Therefore, insects will be able to have more generations per year and, consequently, more crop damage could occur [60–62].

In the insect fauna assessment, two different approaches, based on insect traps, were used. The vinegar bottle traps undifferentiated the capture of a wide range of insects, providing an idea of overall insect diversity, and the Tephri trap, which is specific for *Ceratitis capitata*, is the major plague expected in the vineyard agrosystem.

The vinegar bottle traps captured different groups of insects, with the five major groups being Drosophilidae and Anisopodidae (order Diptera), Nitidulidae and Staphylinidae (order Coleoptera), and Figitidae (order Hymenoptera).

Table 8 shows the seasonal variation in insects captured, using a total of eight vinegar bottle traps. Results are given as total biomass and the total number of insects. Data for the winter season of 2018 were not obtained.

**Table 8.** Quantification of insect diversity of total biomass and number of specimens captured in all seasons.

| | Insect Diversity | | |
|---|---|---|---|
| **Year** | **Season** | **Total Weight (G)** | **Total N° of Specimens** |
| 2018 | Winter | No data | No data |
| | Spring | 0.26 | 123 |
| | Summer | 1.87 | 818 |
| | Autumn | 2.67 | 1123 |
| 2019 | Winter | 0.06 | 25 |
| | Spring | 0.18 | 83 |
| | Summer | 0.68 | 306 |
| | Autumn | 1.47 | 640 |

The summer of 2018 had 2.75-times more insect biomass captured than the same season of 2019 (1.87 g vs. 0.68 g) and the autumn of 2018 had 2.67 g, which represents 1.82-times more biomass than in the same season of 2019, which had 1.47 g of insects (Table 8). Observed wide variation between 2018 and 2019 could be determined by farmers' management or local conditions, including the ones related to climate. Insect ecology and

population dynamics depend on different factors, but it is well known that temperature is one of the most important abiotic factors [63], since insects are ectothermic animals and their body temperature and metabolism depend on the environment temperature [64,65]. Correlation analysis showed a significant positive correlation between insects' biomass and average temperature (r = 0.717 $p \leq 0.05$), maximum temperature (r = 0.763 $p \leq 0.05$), and minimum temperature (r = 0.767 $p \leq 0.05$); for details, see Figure 2. It is well known that higher and lower temperatures have an important role in insect activity and reproduction, beneficial in the first case [60,61], or as a limiting factor for insect activity in the second case [64]. An additional significant positive correlation was obtained with radiation (r = 0.800 $p \leq 0.01$) and negative with soil moisture (r = −0.783, $p \leq 0.05$). This last variable could conditionate the insect's life cycle, during the larval and pupae stage, as shown in the case of several insect species [66,67].

An analysis of all vinegar bottle captures for the 2 years shows that insect captures in winter and spring are characterized by low numbers and are not statistically different. On the other hand, summer and autumn show higher insect density and capture number but are not also statistically different (Figure 8), although summer–autumn seasons are statistically different from winter–spring ones ($p \leq 0.05$). The analysis by year shows that seasonal "insect diversity" has statistical differences ($p \leq 0.05$) between spring and summer–autumn in 2018 and between winter–spring and summer–autumn in 2019. Finally, in 2019, the summer's insect captures were statistically different from other seasons; the same trend was detected between autumn and the remaining seasons ($p \leq 0.05$).

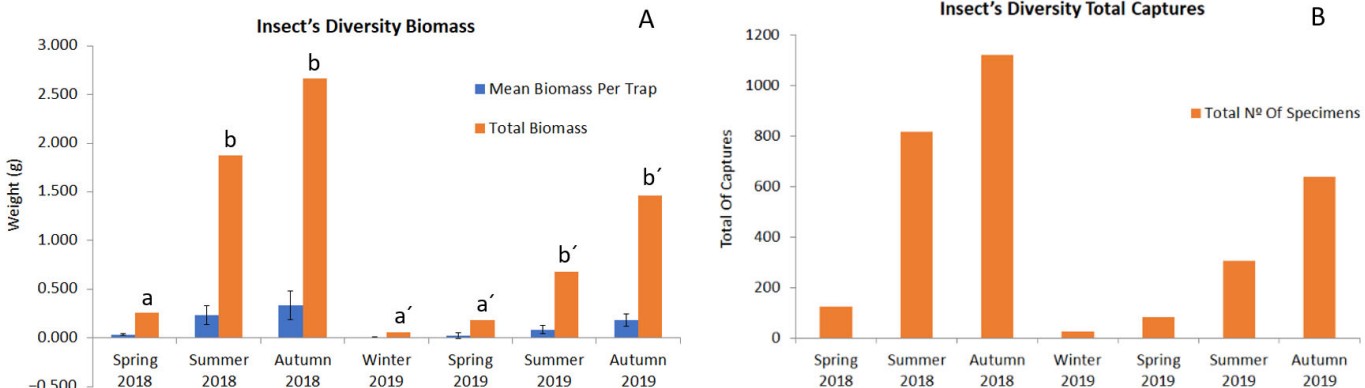

**Figure 8.** Total and mean biomass (**A**) and total captures (**B**) of insect diversity collected by vinegar bottle traps in QV. Different letters indicate a significant difference among them ($p \leq 0.05$). 2018 is represented with the letters a and b, and 2019 with a′ and b′.

The *C. capitata* plague was also monitored and the results are shown as the number of captures through the seasons in Table 9, which also shows the number of captured males and females and the total biomass of the captures. Data for the winter season of 2018 were not obtained.

The total number of specimens ranges from 8 in the winter of 2019 to 1177 specimens in the summer of 2019, representing a 147.13-fold yearly increase in the number of plague specimens captured. When comparing both years, 2019 shows a tendency to have more specimens (and biomass) than 2018, along the seasons. However, in the autumn of 2018, 2.41 more specimens were captured than in 2019 (982 vs. 408 insects) and 2.26 more in terms of biomass (13.35 vs. 5.90 g) (Table 9). A possible explanation is that the higher average and maximum temperature observed in autumn of 2018 (compared with the same season of 2019) could have increased the volatilization of the attractant odors in the trap and, therefore, permit a higher range of action [68,69].

**Table 9.** Quantification of total biomass and number of specimens of *C. capitata* captured in all seasons.

| | | *Ceratitis capitata* | | | |
|---|---|---|---|---|---|
| Year | Season | Total N° of Specimens | Males | Females | Total Weight (g) |
| 2018 | Winter | No data | No data | No data | No data |
| | Spring | 182 | 67 | 115 | 2.22 |
| | Summer | 351 | 175 | 176 | 5.06 |
| | Autumn | 982 | 475 | 507 | 13.35 |
| 2019 | Winter | 8 | 3 | 5 | 0.10 |
| | Spring | 330 | 110 | 220 | 4.51 |
| | Summer | 1177 | 507 | 670 | 17.08 |
| | Autumn | 408 | 105 | 303 | 5.90 |

The summers and autumns are the periods with a larger number of *C. capitata* captured, as expected [68,70], because it overlaps with grapes fructification and fruit ripening. For such a reason, the number of females captured in the summer–autumn period is also higher. Regardless of the similar number of males and females in the summer of 2018 and the winter of 2019, in every season, the number of females is higher than males (Table 9). According to Nolasco and Iannacone [71], referring to other studies, this may be because males, after mating, try to find a new female to copulate, opposite to females that first look for food to be well fed for reproduction and, after mating, look for a substrate to oviposition. Therefore, females were more attracted than males to traps, as they had food attractants. When there are fruits, the effect of attraction is reinforced since the traps are located directly on the vines and near to the fruits.

Figure 9 compares the total biomass of *C. capitata* (A) and the seasonal number of females and males (B) in 2018 and 2019. As in the case of insect diversity (Figure 8), *C. capitata* biomass and captures continuously increase, reaching a maximum in autumn (2018) or summer (2019), which could be related to yearly variation in fruit ripening and higher temperatures in spring of 2019. According to Pimentel [70], it is common for the growth of the insect population activity from July until October/November, when they reach the highest number. This trend was commonly observed for variation in insect diversity and *C. capitata* captures. The highest biomass captures of *C. capitata* recorded in the summer of 2019 could be related to the higher production of grapes in that year, when compared with 2018 (Figure 9). *Vitis vinifera* is a known host of *C. capitata*. [72,73] and the availability of fruits (resources) is a biotic factor directly influencing fruit fly captures and population fluctuation [69,74]. However, the statistical analysis of seasonal *C. capitata* captures only detected significant differences ($p \leq 0.05$) between spring and autumn of 2018. These findings could result from a more stable *C. capitata* population than overall insect diversity.

The analysis of the influence of climate variables in *C. capitata* plague population parameters was also conducted. Similar to insect diversity captures, significant positive correlations between maximum temperature and total biomass (r = 0.712 $p \leq 0.05$) and number of females (r = 0.712 $p \leq 0.05$) were detected. That is expected since, as already stated, the temperature is one of the most important factors regulating insect activity [62] and higher temperatures favor reproduction and development rates [60,61] of this subtropical plague [75]. The same was observed for soil moisture but having negative correlations with total biomass (r = −0.733 $p \leq 0.05$), number of specimens (r = −0.667 $p \leq 0.05$), number of males (r = −0.667 $p \leq 0.05$), and number of females (r = −0.733 $p \leq 0.05$). *C. capitata* has one stage of the life cycle that occurs in the soil, the pupal stage, and soil moisture influences the duration of pupae development and adult emergence, with the second one being higher in drier soils [76]. Specifically, for *C. capitata*, soil temperature had significant positive correlations with total biomass (r = 0.788 $p \leq 0.05$), total of specimens (r = 0.788

$p \leq 0.05$), number of males (r = $-0.788$ $p \leq 0.05$), and total number of females (r = 0.788 $p \leq 0.05$). Soil temperature and soil moisture are the main factors that influence natural mortality and the development of soil-dwelling insect stages [77,78] and they also have an impact on the activity of entomopathogenic microorganisms, such as nematodes, which infect preimaginal stages of *C. capitata* [79].

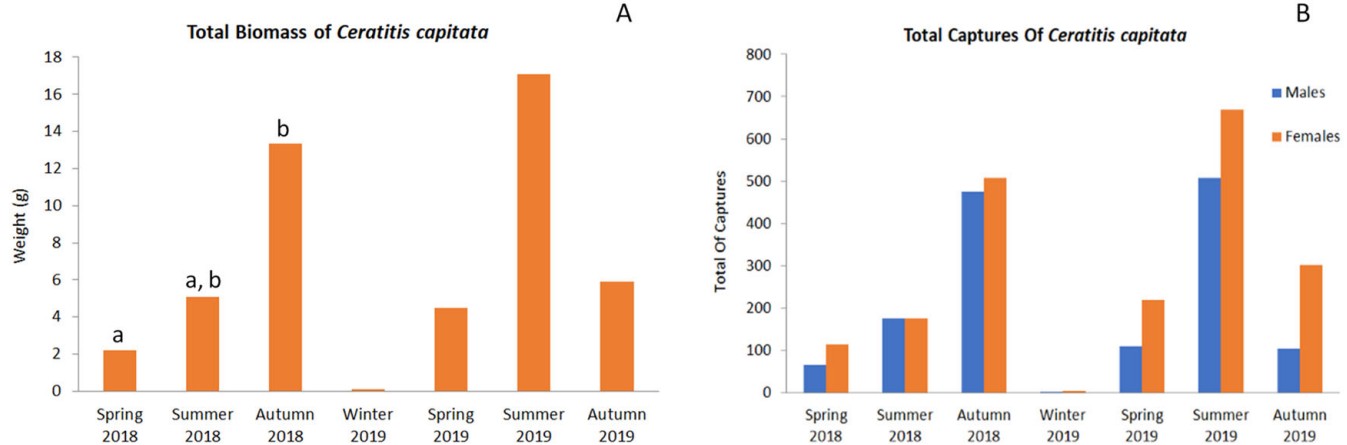

**Figure 9.** Total biomass (**A**) and the total number of males and females (**B**) of *C. capitata* captured in QV. Different letters indicate a significant difference among them ($p \leq 0.05$).

The monitoring of insect diversity to assess the influence of climate was set up based on the hypothesis that climate conditions can promote the *C. capitata* plague and negatively affect the overall insect captures (majorly represented by insects playing important functions in agrosystem services). Although the timeline used is not enough to prove the hypothesis, a trend or pattern seems to be identified. A ratio between overall insect and *C. capitata* captures was calculated, obtaining an average value of 1.34 and varying between 0.25 (less favorable) and 3.13 (more-favorable situation). An imbalance in favor of *C. capitata* plague was observed in the period from spring to autumn, with the exception of summer 2018 and autumn 2019. Overall, these findings seem to support the expected influence of climate, as a strong negative correlation between insect diversity and *C. capitata* indicators was observed.

*3.7. Crop and Production Elements*

The QV is a grapevine farm and an attempt to monitor the influence of climatic conditions on this crop production was made, through the observation of its production cycle, phenological stages, and evaluation of varieties production. The QV agrosystem started to be converted to organic farming production in 2017. In the agrosystem, more than 20 different grapevine varieties and cultivars are grown, because they functioned in a former ampelographic field. Among all these varieties, six of them deserve to be highlighted, e.a. *Malvasia*, *Sercial*, *Terrantez*, *Verdelho*, *Bastardo*, and *Syrah*, which are used in wine production. The first four varieties are usually used to produce Madeira wine and the last ones for staple wine production.

Two production cycles of the individual grapevine varieties were monitored and climatic production data are shown in Tables 10 and 11. The production cycles started in March, with budburst, and ended in September, with vintage, performing a length between 146 and 159 days. The climate data show that the 2019 production cycle was warmer on average 0.7 °C, with a temperature amplitude of 6.5 °C, instead of 6 °C, in 2018, but drier at 7.2 mm. Precipitation of 74.7 mm (2018) and 67.5 mm (2019) was not enough to fulfill crop water requirements, which can reach between 635 and 890 mm, depending on soil and moisture conditions [80,81], and we used 700 mm as average values. The water deficit was calculated at 625.4 (2018) and 632.5 mm in 2019. Starting from 2019, the water deficit

in the QV viticulture was overcome by three periodic irrigations, supplying 900 mm per growth cycle.

**Table 10.** Major climatic conditions and thermic days were observed during the production cycle of grapevine in QV, between the end of March and early September 2018 and 2019.

| Year | Average Temperature (°C) | Maximum Temperature (°C) | Minimum Temperature (°C) | Relative Humidity (%) | Precipitation (mm) | Thermic Days (°C) |
|------|------|------|------|------|------|------|
| 2018 | 19.6 | 22.6 | 16.6 | 72.5 | 74.7 | 2966.3 |
| 2019 | 20.3 | 23.5 | 17.0 | 73.2 | 67.5 | 3025.6 |

**Table 11.** Major grapevine verities production in QV was registered during 2018 and 2019. The table shows the Production Cycle length (PC, days), estimated water deficit, in mm, calculated by the difference between theoretical estimated crop needs and observed precipitation and varieties production in kg per hectare (kg·h$^{-1}$).

| Year | PC Length (days) | Water Deficit (mm) | *Malvasia* (kg·h$^{-1}$) | *Sercial* (kg·h$^{-1}$) | *Terrantez* (kg·h$^{-1}$) | *Verdelho* (kg·h$^{-1}$) | *Bastardo* (kg·h$^{-1}$) | *Syrah* (kg·h$^{-1}$) |
|------|------|------|------|------|------|------|------|------|
| 2018 | 159.0 | 625.4 | 1634.20 | 3989.43 | 1847.52 | 934.47 | 3205.51 | 10,204.08 |
| 2019 | 146.0 | 632.5 | 2695.35 | 4346.10 | 3320.86 | 641.30 | 7817.11 | 16,326.53 |

These climatic conditions allowed for the anticipation of an average production cycle of grapevine varieties in 13 days, because of a higher number of thermic days, 3025.6 °Cd, more 59.3 °Cd than in 2018 (2966.3 °Cd).

From Table 11, among the grape varieties for Madeira wine, the best productivity was obtained by *Sercial*, with 3989.43 and 4346.10 kg·h$^{-1}$, in 2018 and 2019, respectively. Among the grape varieties for the staple wine, *Syrah* shows the highest total production, with 10,204.08 and 16,326.53 kg·h$^{-1}$, in 2018 and 2019, respectively. Comparing all grape varieties, in 2019, they show the following ordination through a decrease in production per hectare: *Syrah* (16,000 kg·h$^{-1}$), *Bastardo* (7817.11 kg·h$^{-1}$), *Sercial* (4346.10 kg·h$^{-1}$), *Terrantez* (3320.86 kg·h$^{-1}$), *Malvasia* (2695.35 kg·h$^{-1}$), and *Verdelho* (641.30 kg·h$^{-1}$). Overall, the local varieties for Madeiran wine show stable productivity, but less than expected. On the other hand, the *Verdelho* variety, even though it is one of the main varieties of Madeira wine, in the analyzed period, showed production volumes well below expectations. Variety production can be improved with the acquisition of a localized irrigation system, according to the work of Fraga et al. [81], showing an increase in vine production with the introduction of drip irrigation. Given the immense number of vine varieties in the study area, it can be concluded that some are better adapted to the climatic conditions present in the study area than others, a fact that can easily be observed in the volumes of the final productions obtained.

## 4. Final Remarks

During the 2 years of data recording in *Quinta das Vinhas*, if we consider the summer and winter seasons as representatives of the climatic extreme events, then significant differences between average and maximum average temperatures were observed, which follows the data from recent meteorological series, but not for minimum average temperatures. The same significant differences were found between these two seasons for average precipitation, ETP, soil humidity, and water deficit. This shows that QV represents an example of an agrosystem, located in Madeira, with clear climatic differences between these two seasons, leading to differences in edaphic properties, microbial, floristic, and insect diversity. These findings provide insights into the consequences of extremely high temperature and low precipitation events that can be soon more common in other parts of the island. In the

2-year data (2018–2019), medium and maximum average temperatures in summer were 5.2 and 3.8 °C higher than in winter, while precipitation was 89% lower, which significantly impacted soil humidity, ETP, and water deficit. If we consider summer and winter extreme events as proxies for future climate impacts, we can observe that discrepancy between these two seasons in QV is far higher than the projected average differences between current and future temperatures in the Climate—Madeira Strategy [5].

The observed changes in floristic and insect composition in this study case, as well as soil characteristics, can give us an idea of what to expect in general on the island agrosystems in the future. It has already been discussed how the microorganism groups seem to show few significant differences between summer and winter events, presenting some stability within the communities, although with reduced numbers in these seasons. Higher abundance and diversity of microorganisms were found in spring and autumn, suggesting a better environment for microbial communities. These results show the detrimental effects that extreme weather, including forecasted higher temperatures and reduced precipitation, could have, even in these communities. Our study also seems to identify a critical imbalance between nitrogen-fixing and denitrifying bacteria, especially in summer, which could be determined by the rise in temperature and drought. Regarding the floristic surveys, a significant difference between summer and winter was observed in the number of individuals, as well as in the number of therophytes, representing the most abundant group of species in this agrosystem. These floristic indicators presented significant negative correlations with temperature and for the number of individuals, a significant correlation was also found with soil humidity. It is known that precipitation patterns as well as ETP play a fundamental role in shaping floristic indicators [82–85]. We can then extrapolate that increased temperatures and reduced precipitation forecasted will lead to reduced floristic diversity and seasonal cover in the island agrosystems. The captured insect's biomass, *C. capitata* biomass, and the total number of individuals showed a significant difference between summer–autumn and winter–spring. Although other factors, such as the increase in fruit production, impact overall numbers, it seems that, for this agrosystem, the rise in the summer period temperature and the decrease in precipitation favors the development of *C. capitata* and other insect populations. In the future, this can be presented as a problem, since it could lead to an increase in plague-related losses to Madeira's agriculture. From the production of the six most prominent grape varieties in QV, we can observe that five of them show an increase in 2019, which can be explained, according to our data, by the fact that temperatures were moderately higher and more uniform in 2019 compared to 2018. On the other hand, 2019 had much less precipitation than 2018. The expected temperature rise could have a strong impact on fruit productivity, while in those 2 years, irrigation probably offset the loss of precipitation impacts.

Data obtained from the agrosystems' indicators provide a baseline as a starting point for long-term monitoring and allow for further quantifying the influence of climate changes on agrosystem productivity, resilience, and sustainability. Notwithstanding, this monitoring is intended to be carried out for a longer period, providing more accurate information and contributing to the robustness of the model.

**Author Contributions:** Conceptualization, M.C.O.O., J.F.T.G., F.L.M., J.G.R.d.F., H.N. and M.Â.A.P.d.C.; investigation, C.R., M.C.O.O., F.R., F.L.M. and H.N.; writing—original draft preparation, C.R., M.C.O.O., F.R., F.L.M., J.F.T.G., H.N. and M.Â.A.P.d.C.; writing—review and editing, C.R., J.F.T.G. and M.Â.A.P.d.C.; supervision, M.Â.A.P.d.C.; funding acquisition, J.F.T.G., J.G.R.d.F. and M.Â.A.P.d.C. All authors have read and agreed to the published version of the manuscript.

**Funding:** This research was funded by *Programa Operacional Madeira* 14–20, Portugal 2020, and the European Union through the European Regional Development Fund, grant number M1420-01-0145-FEDER-000011 [CASBio] and by the *Agência Regional para o Desenvolvimento da Investigação, Tecnologia e Inovação*, Portugal 2020 and the European Union through the European Social Fund [grant number M1420-09-5369-FSE000002, ARDITI].

**Data Availability Statement:** Not applicable.

**Acknowledgments:** The authors acknowledge the support by National Funds FCT-Portuguese Foundation for Science and Technology, under the projects UIDB/04033/2020 and UIDP/04033/2020. The *Instituto Português do Mar e da Atmosfera* (IPMA), for the availability of historical climate data; the APOGEO project through the Cooperation Program INTERREG-MAC 2014–2020, for imaging; to the *Quinta Das Vinhas Madeira* for the study site availability; and the *Secretarias Regionais de Agricultura e Desenvolvimento Rural e Recursos Naturais e Alterações Climáticas* for partnership and support.

**Conflicts of Interest:** The authors declare no conflict of interest.

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
