# Peer review of "Anticipating the Climate Change Impacts on Madeira’s Agriculture: The Characterization and Monitoring of a Vine Agrosystem"

_agronomy, doi:10.3390/agronomy12092201_

Round 1
Reviewer 1 Report
This manuscript is technically sound, generally well-written, methods, and analysis. The results are fairly clearly presented. However, there are several factors which detract from the quality of the manuscript. The reviewer cannot recommend publication of this work in its present form. Therefore, the manuscripts should have the minor revision. The reviewer lists some comments and suggestions for improvements below.
1. Please rewrite the abstract as follows: 1-2 sentences on the context and the need for the study; several sentences on the model; 2-3 sentences on how the model can be applied and its capabilities; 1-2 sentences on key conclusions and recommendations.
2. The abstract should include quantitative results
3. The current Introduction is too simple, it should include background, current progress, research gaps and the objective of this study, etc (Please emphasize the novelty and impactful contribution of this work as currently this appears to be marginal. The scientific contributions of this study could be further improved).
4. For readers to quickly catch your contribution, it would be better to highlight major difficulties and challenges, and your original achievements to overcome them, in a clearer way in abstract and introduction.
5. Full names should be shown for all abbreviations in their first occurrence in texts.
6. Please cite the corresponding references for the models/equations/formulas that were not originally developed by yourself.
Author Response
Response to Reviewer 1 Comments
Manuscript ID: agronomy-1899536
Type of manuscript: original article
Title: Anticipating the climate changes impacts on Madeira´s agriculture: the characterization and monitoring of a vine agrosystem
Authors: Miguel Â. A. Pinheiro de Carvalho, Carla Ragonezi, Maria Cristina O. Oliveira, Fábio Reis, Fabrício Lopes De Macedo, José G. R. Freitas, Humberto Nóbrega, José Filipe T. Ganança.
The authors appreciate the reviewer’s comments about the manuscript. The comments were addressed properly below and can be visualized in the resubmitted version.
Point 1: Please rewrite the abstract as follows: 1-2 sentences on the context and the need for the study; several sentences on the model; 2-3 sentences on how the model can be applied and its capabilities; 1-2 sentences on key conclusions and recommendations.
Response 1: The authors appreciate the reviewer´s comment and proceed to the correction in the abstract (please see the revised version).
Point 2: The abstract should include quantitative results
Response 2: The authors appreciate the reviewer´s comment and proceed to add the information in the abstract (please see the revised version).
Point 3: The current Introduction is too simple, it should include background, current progress, research gaps and the objective of this study, etc (Please emphasize the novelty and impactful contribution of this work as currently this appears to be marginal. The scientific contributions of this study could be further improved).
Response 3: The authors appreciate the reviewer´s comment and proceed to add references in the introduction section (please see the revised version).
Point 4: For readers to quickly catch your contribution, it would be better to highlight major difficulties and challenges, and your original achievements to overcome them, in a clearer way in abstract and introduction.
Response 4: The authors appreciate the reviewer´s comment and proceed to the correction in the abstract and introduction (please see the revised version).
Point 5: Full names should be shown for all abbreviations in their first occurrence in texts.
Response 5: The authors appreciate the reviewer´s comment and proceed to the correction throughout the article (please see the revised version).
Point 6: Please cite the corresponding references for the models/equations/formulas that were not originally developed by yourself.
Response 6: The authors appreciate the reviewer´s comment and proceed to add the information in the methodology and reference sections (please see the revised version).
Reviewer 2 Report
Very good research, with a great network of information and very good handling of the data.
Only two points:
1) You should specify a clearer and more concise objective. Monitoring and surveying must be linked to the models to be evaluated in order to give meaning to the work.
2) For non-normal data, Spearman's correlation should be used.
Author Response
Response to Reviewer 2 Comments
Manuscript ID: agronomy-1899536
Type of manuscript: original article
Title: Anticipating the climate changes impacts on Madeira´s agriculture: the characterization and monitoring of a vine agrosystem
Authors: Miguel Â. A. Pinheiro de Carvalho, Carla Ragonezi, Maria Cristina O. Oliveira, Fábio Reis, Fabrício Lopes De Macedo, José G. R. Freitas, Humberto Nóbrega, José Filipe T. Ganança.
The authors appreciate the reviewer’s comments about the manuscript. The comments were addressed properly below and can be visualized in the resubmitted version.
Point 1: You should specify a clearer and more concise objective. Monitoring and surveying must be linked to the models to be evaluated in order to give meaning to the work.
Response 1: The authors appreciate the reviewer´s comment and proceed to the alteration in the objective (please see the end of the introduction section in the revised version).
Point 2: For non-normal data, Spearman's correlation should be used.
Response 2: The authors appreciate the reviewer´s comment and proceed to the alteration adding the data in the results and discussion section in the correlations analysis.